# Ctrl-Z Sampling: Scaling Diffusion Sampling with Controlled Random Zigzag Explorations

## Abstract

Diffusion models have shown strong performance in conditional generation by progressively denoising Gaussian samples toward a target data distribution. This denoising process can be interpreted as a form of hill climbing in a learned representation space, where the model iteratively refines a sample toward regions of lower noise and higher quality. However, this learned climbing often converges to local optima with plausible but suboptimal generations due to latent space complexity and suboptimal initialization. While prior efforts often strengthen guidance signals or introduce fixed exploration strategies to address this, they exhibit limited capacity to escape steep local maxima. In contrast, we propose Controlled Random Zigzag Sampling (`Ctrl-Z Sampling`), a novel sampling strategy that adaptively detects and escapes such traps through controlled exploration. In each diffusion step, we monitor the trajectory of quality-scores of predictions over denoising steps, given a reward model that serves as a surrogate for the underlying sample quality, and identify plateaus as local optima along this trajectory. Upon such detection, we inject noise and revert to a previous, noisier state to escape the current plateau. The reward model then evaluates candidate trajectories, accepting only those that offer improvement, otherwise scheming progressively deeper explorations when nearby alternatives fail. This controlled zigzag process allows dynamic alternation between forward refinement and backward exploration, enhancing both alignment and visual quality in the generated outputs. The proposed method is model-agnostic and also compatible with existing diffusion frameworks. Experimental results show that `Ctrl-Z Sampling` consistently improves generation quality across different NFE budgets compared to the original sampler.

## 1 Introduction

Diffusion models have emerged as a powerful class of generative models, achieving state-of-the-art performance across a wide range of generative tasks, including image (Rombach et al., 2022; Podell et al., 2023), video (Ho et al., 2022; Blattmann et al., 2023), text (Li et al., 2022), audio (Kong et al., 2021), 3D object (Poole et al., 2023), along with a growing number of domains (Chen et al., 2024a; Xu et al., 2022; Chamberlain et al., 2021; Janner et al., 2022). These models operate by gradually transforming Gaussian noise into data samples through an iterative denoising process (Sohl-Dickstein et al., 2015; Ho et al., 2020; Song et al., 2021). Modern variants are typically designed to accept conditional inputs, enabling guided generation based on class labels, texts, or other structured signals (Zhang et al., 2023; Bansal et al., 2023).

Despite their strong generative performance, diffusion models often exhibit semantic misalignment or global inconsistency in conditional generation. Taking image generation as an example, the denoising process is an iterative refinement procedure on the data manifold. In practice, this refinement can stagnate in suboptimal regions where the current sample already looks locally plausible but remains semantically flawed (*e.g.*, missing objects, incorrect relations, or anatomically implausible details). We refer to this sampling stagnation or collapse as converging to a *local optima* in a conceptual sample quality space, where quality reflects how faithful, structurally correct, and well aligned the image is with the text condition. Such failures are commonly triggered by misaligned initial

noise, which misguides the trajectory toward outputs that are visually compelling but misaligned with the intended condition (Xu et al., 2024).

Some existing works attempt to escape local maxima in the diffusion process by reinforcing conditional information—either through classifier-free guidance (CFG) (Ho & Salimans, 2022), which amplifies the conditioning signal during denoising, or through repeated conditional denoising and unconditional inversion (Lugmayr et al., 2022; Bai et al., 2024), which explicitly trade off visual plausibility to explore trajectories more faithful to the input. While these methods enhance the influence of input conditions, the way conditional information is injected remains limited and loosely controlled, often leading to suboptimal or ineffective guidance during generation. Similarly, recent test-time scaling approaches propose mutating the current latent state or stepping backward to a previous timestep (Ma et al., 2025; He et al., 2025; Zhang et al., 2025c), introducing stochasticity to escape local maxima under the guidance of reward models. Although these methods can identify better-aligned alternative states and diffusion trajectories in principle, they often require evaluating a large number of candidate states to locate promising alternatives. This renders them less practical in local inference settings, and the fixed extent of mutation or rollback may still leave the model trapped when the maxima are broad or steep.

In this paper, we propose Controlled Random Zigzag Sampling (`Ctrl-Z Sampling`), a novel diffusion sampling strategy that enhances diffusion generation by performing backward searches through randomized explorations into adaptively noisier latent manifolds. This enables the model to escape broad local maxima through flexible control of escape strength, governed by the degree of noise inversion. Specifically, `Ctrl-Z` Sampling detects potential local maxima based on a stagnation criteria in the preference scores assigned by a reward model, which we use as a surrogate for the conceptual sample quality space. Upon detection, it injects noise and reverts to an earlier, noisier timestep to encourage exploration beyond the current optimization plateau. The reward model then evaluates each candidate state, accepting only those that yield more promising future predictions. When no improvement is found nearby, the model continues retreating to deeper noise levels, increasing the likelihood of escape. This process forms a controlled random zigzag trajectory that dynamically alternates between forward refinement and backward exploration, ultimately improving both condition alignment and overall generation quality.

`Ctrl-Z` Sampling is broadly applicable to different diffusion models. Experimental results on text-to-image benchmarks show that `Ctrl-Z` Sampling achieves superior generation quality across multiple metrics, with only $6.72\times$ more function evaluations compared to standard sampling. This demonstrates its potential as a practical alternative to other expensive test-time scaling methods, particularly in local inference settings. To summarize, our key contributions are:

- We frame conditional diffusion generation as a hill-climbing process in the sample quality space, and show that existing sampling strategies may become trapped in broad local maxima that exhibit conditional misalignment or global inconsistency, due to limited escape strength.

- We propose Controlled Random Zigzag Sampling (`Ctrl-Z` Sampling), a novel diffusion sampling strategy that enables controlled escape from local maxima through reward-guided exploration. By injecting noise with adaptive strength upon detecting stagnation, `Ctrl-Z` Sampling selectively reverts to higher-noise states to explore alternative paths, allowing the model to overcome both narrow and broad local optimal during generation.

- We conduct extensive experiments demonstrating that `Ctrl-Z` Sampling substantially improves text-to-image generation quality. Experiments show that, unlike search methods relying on numerous shallow trials across many directions, `Ctrl-Z` Sampling achieves better exploration efficiency by making fewer, progressively deeper steps in the generation landscape to escape local maxima.

## 2 RELATED WORKS

**Diffusion Models Inference Techniques.** Diffusion models generate data by gradually transforming Gaussian noise into structured samples through iterative denoising (Sohl-Dickstein et al., 2015; Ho et al., 2020; Song et al., 2021), with both U-Net-based (Rombach et al., 2022; Podell et al., 2023; Saharia et al., 2022) and Transformer-based architectures (Peebles & Xie, 2023; Li et al.,

2025a) achieving strong results across modalities. However, training or finetuning remains costly, and constrained by computational demands (Wallace et al., 2024; Fan et al., 2023). To address these challenges, various inference-time techniques have been proposed to enhance conditional generation (Dhariwal & Nichol, 2021). Classifier-free guidance (CFG) (Ho & Salimans, 2022; Chung et al., 2024) contrasts conditional and unconditional predictions to improve alignment, though at the risk of fidelity loss. Others guide generation by modifying attention maps during sampling (Feng et al., 2023; Chefer et al., 2023; Li et al., 2023; Hong, 2024), but these can struggle with abstract or global conditions. While pruning techniques for diffusion models facilitate efficient inference by reallocating saved compute to additional denoising steps for enhanced generation quality (Fang et al., 2023; Li et al., 2024a; Seo et al., 2025), they do not improve per-step denoising quality independently. More recent methods exploit the sensitivity of generation to noise initialization, leveraging information leakage from noisy priors (Lin et al., 2024; Everaert et al., 2024) and developing strategies to optimize or select noise vectors (Xu et al., 2024; Mao et al., 2023; Zhou et al., 2024; Guo et al., 2024; Samuel et al., 2024) (Tang et al., 2024), or inject low-frequency condition cues (Guttenberg, 2023). However, guided optimization of noise latents can be inefficient and may also lead to degenerate solutions by collapsing the denoising process. In contrast, `Ctrl-Z` Sampling improves conditional alignment without retraining or costly optimization by adaptively enhancing exploration strength, enabling escape from broad local maxima and improving generation quality at inference.

**Diffusion Sampling Strategies.** DDPM (Ho et al., 2020) introduces diffusion sampling as a stochastic reverse process, which DDIM (Song et al., 2021) later reformulates as a deterministic ODE for faster inference. Building on this, recent methods such as DPM-Solver (Lu et al., 2022), UniPC (Zhao et al., 2023), and AYS (Sabour et al., 2024) improve both efficiency and quality through high-order ODE solvers and adaptive step sizes, while some other work incorporates directional guidance (Watson et al., 2022; Ghosh et al., 2023; Choi et al., 2024). Later works improve sampling quality via latent-space inversion or zigzag explorations (Lugmayr et al., 2022; Xu et al., 2023b; Bai et al., 2024). In parallel, guided search by reward models has been explored via particle-based sampling (Kim et al., 2025; Li et al., 2025b) and beam search (Oshima et al., 2025). More recent efforts extend these toward test-time scaling in diffusion models (Ma et al., 2025; Singhal et al., 2025; He et al., 2025; Zhang et al., 2025c), aiming to effectively improve sampling performance with more function evaluations. However, these approaches primarily focus on enlarging the candidate pool through shallow, local perturbations, often neglecting the role of exploration depth and thus remaining vulnerable to local optima. In contrast, concurrent works (Zhang et al., 2025c; Jain et al., 2025) adopt DFS-style search strategies that roll back diffusion states to significantly earlier, noisier timesteps, increasing exploration depth but requiring costly re-denoising from these deeper noise levels. In contrast, `Ctrl-Z` Sampling mitigates this by dynamically intensifying exploration into higher-noise manifolds only when stagnation is detected, allowing deeper search precisely where needed and avoiding unnecessary computation elsewhere.

## 3 PRELIMINARIES

**Diffusion Process.** Diffusion models are generative frameworks that synthesize data by progressively denoising a sample drawn from a Gaussian prior. Let $\mathcal{N}$ represent the standard normal distribution and $\mathcal{D}$ denote the target data distribution. Starting from a noise sample $x_T \sim \mathcal{N}$, a sample $x_0 \in \mathcal{D}$ is generated through a sequence of $T$ denoising steps, conditioned on $x_T$ and auxiliary prompt $c$: $x_0 = \Phi(x_T, c)$, where $\Phi$ denotes the sampling procedure, which may be stochastic or deterministic.

**Deterministic Sampling with DDIM.** We adopt the DDIM sampling framework (Song et al., 2021), which enables efficient and deterministic generation by removing stochasticity from the denoising process. Given a predefined noise schedule $\{\beta_t\}_{t=1}^T$, let $\alpha_t = \prod_{i=1}^t (1 - \beta_i)$ denote the cumulative product. The generation process is decomposed into $T$ successive mappings:

$$x_0 = \Phi^1 \circ \Phi^2 \circ \cdots \circ \Phi^T(x_T, c), \tag{1}$$

where each mapping $\Phi^t$ transforms $x_t$ into $x_{t-1}$ using the predicted noise $\epsilon_\theta^t(x_t, c)$:

$$x_{t-1} = \Phi^t(x_t, c) = \sqrt{\alpha_{t-1}} \cdot \frac{x_t - \sqrt{1 - \alpha_t} \cdot \epsilon_\theta^t(x_t, c)}{\sqrt{\alpha_t}} + \sqrt{1 - \alpha_{t-1}} \cdot \epsilon_\theta^t(x_t, c). \tag{2}$$

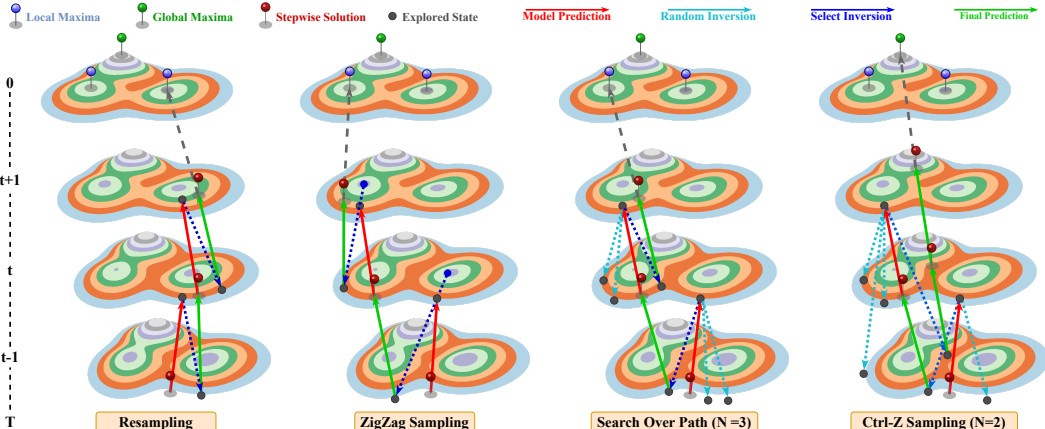

Figure 1: Illustration of different sampling strategies during the diffusion process. The diffusion process is illustrated as ascending a rugged landscape for sample quality and condition alignment, where each strategy searches for improved solutions while avoiding traps at local maxima. Each strategy begins with a denoising step along the predicted direction (**red**), followed by either a single or multiple inversion explorations. The selected/executed inversions are marked in **blue**, while discarded ones are in **cyan**. The accepted inversion is followed by a forward denoising step (**green**). Subsequent steps are omitted and shown in **gray**. Unlike prior methods, the proposed `Ctrl-Z` Sampling adaptively increases exploration strength through larger inversion steps when local perturbations fail to reveal improved trajectories, enabling escape from broader local maxima. $N$ denotes inversion candidates per exploration stage.

Note the $\sigma$ term in DDIM is zeroed and omitted for determinism. The first term in Equation (2) approximates the clean sample corresponding to $x_t$, which we denote as:

$$\hat{x}_0^{t-1}(x_t, c) = \frac{x_t - \sqrt{1 - \alpha_t} \cdot \epsilon_\theta^t(x_t, c)}{\sqrt{\alpha_t}}. \tag{3}$$

This intermediate estimate serves as a useful proxy for the final output and is used in our method to assess semantic alignment via a reward model. Although $\Phi^t$ formally returns only $x_{t-1}$, we compute and reuse $\hat{x}_0^{t-1}$ jointly for efficiency.

**Latent Inversion.** DDIM defines a deterministic denoising trajectory but lacks an explicit inversion re-noising mechanism. To reintroduce noise in a controlled manner, we define an inversion operator $\Psi(x_t, \Delta)$, which simulates a forward transition from $x_t$ to a noisier latent state $x_{t+\Delta}$ using:

$$x_{t+\Delta} = \Psi(x_t, \Delta; \epsilon) = \sqrt{\frac{\alpha_{t+\Delta}}{\alpha_t}} \cdot x_t + \sqrt{1 - \frac{\alpha_{t+\Delta}}{\alpha_t}} \cdot \epsilon, \tag{4}$$

where $\alpha_t = \prod_{i=1}^{t}(1 - \beta_i)$ is the cumulative noise schedule inherited from the original diffusion process, and $\epsilon \sim \mathcal{N}(0, \mathbf{I})$ a random noise.

This operator enables transitions into higher-noise manifold of the latent space while retaining denoising progress. It provides a mechanism for test-time exploration that perturbs the generation trajectory without completely discarding previously accumulated semantic structure.

## 4 METHODS

In this section, we discuss the intuition behind `Ctrl-Z` Sampling compared to other sampling techniques in the context of a diffusion hill climbing scenario in text-to-image generation, and illustrate the technical details of `Ctrl-Z` Sampling. A full algorithm of our method is given in Alg. 1.

**Denoising As A Hill Climbing Problem.** As illustrated in Fig. 1, we focus on deterministic diffusion sampling, which we view as hill climbing in latent space with respect to an abstract notion of sample quality and condition alignment. Conceptually, we posit an abstract score functional

$Q(x, c)$ that measures the general quality an sample $x$ for condition $c$. Because $Q$ is not directly observable, we approximate it with off-the-shelf reward models $R(x, c)$ that scores sample quality or condition alignment, which serve as surrogates for this conceptual quality space. During sampling, each denoising step produces a clean estimate $\hat{x}_0^{(t)}$ and a scalar reward $R(\hat{x}_0^{(t)}, c)$, yielding a one-dimensional prediction-quality trajectory $\{R(\hat{x}_0^{(t)}, c)\}_t$ over timesteps. However, this trajectory often settles into *local optima*: regions where the sample is plausible with high prior probability from $\Phi$, but misaligned with low reward based on $R$. In this view, the standard denoising trajectory represents a greedy optimization path that lacks the ability to escape the peak of attraction of these suboptimal states (as visualized in the trajectory analysis in Fig. 7).

In the illustrated example, Resampling performs a single-step random inversion to perturb the latent state and explore nearby alternatives, but such exploration is limited in scope and may fail to yield meaningful improvements. Zigzag Sampling alternates between a backward inversion step along the unconditional trajectory and a forward, text-conditioned denoising step, so that the prompt conditioning signal is repeatedly re-injected into the sampling path. Search-over-path (SOP) extends Resampling by evaluating multiple perturbed candidates, but its search depth is fixed, limiting its ability to escape broader local maxima. In contrast, the proposed Controlled Random Zigzag Sampling (`Ctrl-Z` Sampling) adaptively adjusts the strength of latent perturbation based on reward feedback. When shallow exploration fails, `Ctrl-Z` Sampling incrementally expands its search radius, enabling escape from both narrow and broad local maxima while preserving output quality.

**Controlled Random Zigzag Sampling.** To enable conditional diffusion models to escape local optimal, we propose a reward-guided sampling strategy that incorporates controlled backward exploration. Our approach detects potential sub-optimal states using a reward model, then dynamically perturbs the latent trajectory (*i.e.*, the sequence of intermediate latent states $x_t$ produced by DDIM sampling) through guided inversion steps of increasing strength. This process forms a zigzag trajectory in the latent space, alternating between conditional refinement and noise-space exploration.

---

**Algorithm 1** `Ctrl-Z` Sampling

1: **Input:** Denoising operator $\Phi^t$, clean sample estimate $\hat{x}_0^{t-1}(x_t, c)$, inversion operator $\Psi(x_t, \Delta; \epsilon)$, condition $c$, total steps $T$, reward model $R$, exploration window $\lambda$, accept threshold $\delta$, max inversion depth $d_{\max}$, max number of candidates $N$.
2: **Output:** Final image estimate $x_0$
3: Sample Gaussian noise $x_T$
4: Initialize reward score $r_{\text{prev}} \leftarrow -\infty$
5: **for** $t = T$ to $1$ **do**
6:     $x_{t-1} \leftarrow \Phi^t(x_t, c); \quad \hat{x}_0^{t-1} \leftarrow \hat{x}_0^{t-1}(x_t, c)$    # Eq. (2, 3)
7:     **if** $t > T - \lambda$ **then**
8:         $r \leftarrow R([c], \hat{x}_0^{t-1})$
9:         **if** $r \geq r_{\text{prev}} + \delta$ **then**
10:             $r_{\text{prev}} \leftarrow r$
11:         **else**
12:             inversion_step $\leftarrow 1$, best_score $\leftarrow r$, best_state $\leftarrow x_{t-1}$
13:             **while** inversion_step $\leq d_{\max}$ **do**
14:                 $\Delta \leftarrow \min(\text{inversion\_step}, T - t)$
15:                 **for** $i = 1$ to $N$ **do**
16:                     Sample $\epsilon \sim \mathcal{N}(0, \mathbf{I})$
17:                     $\tilde{x}_{t+\Delta} \leftarrow \Psi(x_t, \Delta; \epsilon)$    # Inversion step
18:                     **for** $k = t + \Delta$ to $t$ **do**
19:                         $\tilde{x}_{k-1} \leftarrow \Phi^k(\tilde{x}_k, c)$
20:                     **end for**
21:                     Estimate $\hat{x}_0^{t-1} \leftarrow \hat{x}_0^{t-1}(\tilde{x}_t, c)$
22:                     $r_{\text{cand}} \leftarrow R(c, \hat{x}_0^{t-1})$
23:                     **if** $r_{\text{cand}} > \text{best\_score}$ **then**
24:                         best_score $\leftarrow r_{\text{cand}}$,    best_state $\leftarrow \tilde{x}_{t-1}$
25:                     **end if**
26:                 **end for**
27:                 **if** best_score $\geq r_{\text{prev}} + \delta$ **then**
28:                     **break** from search
29:                 **end if**
30:                 inversion_step $\leftarrow$ inversion_step $+ 1$
31:             **end while**
32:             $x_{t-1} \leftarrow \text{best\_state}, \quad r_{\text{prev}} \leftarrow \text{best\_score}$
33:         **end if**
34:     **end if**
35:     $x_t \leftarrow x_{t-1}$
36: **end for**
37: **return** $x_0$

---

**Local Maxima Detection.** At each selected step $t$, the clean sample estimate $\hat{x}_0^{t-1}$ is computed from the current latent state $x_t$ using Equation (3), and its quality is evaluated by a reward model $R(c, \hat{x}_0^{t-1})$ that assigns a scalar score to the candidate state. A local maximum is detected when the current reward falls below the previous best by more than a threshold:

$$R(c, \hat{x}_0^{t-1}) < r_{\text{prev}} + \delta, \tag{5}$$

where $\delta$ denotes the acceptance threshold. By default, $\delta = 0$ enforces non-decreasing reward over time. $\delta$ governs the minimal improvement required for a candidate to be accepted. A positive $\delta$ suppresses trivial gains and promotes robustness in candidate selection, while $\delta \leq 0$ allows more permissive updates, including those with marginal or slightly lower reward values, which may carry

a higher risk of degraded outputs. This criterion provides a simple yet effective signal for identifying local maxima during sampling. However, relying on a fixed offset from the prior best can rigidly penalize high-reward plateaus, especially when $\delta$ is large. Designing more flexible criteria such as relative thresholds or global reward schedulers could mitigate such cases and improve adaptability. We leave such extensions to future work.

**Controlled Latent Inversion.** To escape from a detected local maximum, we apply the inversion operator $\Psi(x_t, \Delta; \epsilon)$ (defined in Equation (4)) to perturb the current latent $x_t$ toward a higher-noise state $x_{t+\Delta}$. The step size $\Delta \in \mathbb{N}$ corresponds to discrete timestep indices in the DDIM schedule, where smaller values induce mild perturbations and larger values enable broader exploration. This controlled noise injection serves as a test-time search mechanism that reintroduces uncertainty while preserving structural information acquired during prior denoising, thereby enabling the model to explore nearby alternative trajectories effectively.

**Candidate Selection and Adaptive Inversion Step.** After applying the inversion, the model resumes forward denoising from $x_{t+\Delta}$ to $x_{t-1}$ using standard conditional DDIM steps $\Phi^k(\cdot, c)$ for $k = t + \Delta, \ldots, t$. The resulting candidate $\hat{x}_0^{\,t-1}$ is then evaluated by the reward model. At each iteration, $N$ candidates are generated using distinct noise vectors $\epsilon$, and the one with the highest reward is selected. If its score exceeds the most recently accepted reward by at least a threshold $\delta$, the update is accepted.

Otherwise, the inversion step size $\Delta$ is incrementally increased (*e.g.*, from 1 to $d_{\max}$), enabling progressively stronger perturbations to search for better trajectories. This defines a greedy yet adaptive hill-climbing strategy: the search proceeds until a satisfactory candidate is found or the maximum inversion depth is reached. If no improvement beyond the threshold is identified, the best candidate observed during the search is retained, promoting stability while often yielding results comparable to or better than the default forward step.

The inversion depth can be unbounded with $d_{max} = \infty$, and the candidate budget $N$ per step guarantees termination. The candidate evaluation process is inherently parallelizable since each candidate trajectory is conditionally independent until selection, thus can be efficiently distributed across computational resources, supporting scalable inference with minimal overhead.

Empirically, at early timesteps (*e.g.*, $t = 600/1000$), the DDIM update already assigns substantial weight (above 0.02) to the clean estimate $\hat{x}_0^{\,t-1}$ via Equation (2), meaning $x_{t-1}$ is strongly influenced by $\hat{x}_0^{\,t-1}$. As a result, much of the image's low-frequency structure is established early, making later corrections less effective. Therefore, `Ctrl-Z` Sampling restricts exploration to the first $\lambda < T$ denoising steps, where trajectory perturbations can meaningfully influence the final output, whereas updates at larger $\lambda$ often encounter harder-to-escape plateaus and yield diminishing returns.

## 5 EXPERIMENTS AND RESULTS

### 5.1 EXPERIMENT SETTINGS

We evaluate our method on three representative text-to-image benchmarks: **Pick-a-Pic** (Kirstain et al., 2023), **DrawBench** (Saharia et al., 2022), and **T2I-CompBench** (Huang et al., 2023) covering real-world diversity, compositional complexity, and large-scale open-world scenarios. Evaluations are conducted using four metrics: **HPSv2** (Wu et al., 2023), **PickScore** (Kirstain et al., 2023), **ImageReward (IR)** (Xu et al., 2023a), and **Aesthetic Score (AES)** (Schuhmann et al., 2022). All experiments are conducted on SD 2.1 base (Feng et al., 2023) and Hy-DiT (Li et al., 2024b). We use $T = 50$ inference steps with CFG (scale 5.5), an exploration window $\lambda = 40$, $N = 4$ candidates, $d_{\max} = 3$, $\delta = 0$, and ImageReward as the reward model. We compare `Ctrl-Z` **Sampling** against standard **DDIM**, **Resampling** (Lugmayr et al., 2022), **Z Sampling** (Bai et al., 2024), and **SOP** (Ma et al., 2025) under settings with comparable number of function evaluations (NFEs), which is reported as the average number of denoiser forward passes *per denoising step* (rather than the total per generation). More details are available in the Appendix Section B.

Table 1: Quantitative results of sampling methods on Pick-a-Pic and DrawBench, evaluated using four human-aligned metrics. Results are reported for both Stable Diffusion 2.1 and Hunyuan-DiT, along with average number of function evaluations (NFEs). † and ‡ indicate `Ctrl-Z` variants with different exploration parameters, showing performance under varying NFEs.

| Method | Pick-a-Pic↑ | | | | DrawBench↑ | | | | NFEs↓ |
|---|---|---|---|---|---|---|---|---|---|
| | HPS v2 | AES | PickScore | IR | HPS v2 | AES | PickScore | IR | |
| **Stable Diffusion 2.1** | | | | | | | | | |
| DDIM | 25.34 | 5.649 | 20.67 | 0.194 | 24.90 | 5.410 | 21.39 | 0.046 | 1.00 |
| Resampling | 26.00 | 5.648 | 20.78 | 0.322 | 25.49 | 5.385 | 21.52 | 0.205 | 2.00 |
| Z-Sampling | 26.29 | 5.653 | 20.71 | 0.346 | 25.95 | 5.465 | 21.47 | 0.296 | 3.00 |
| SOP-1 | 26.34 | 5.684 | 20.85 | 0.735 | 25.78 | 5.426 | 21.64 | 0.637 | 3.00 |
| SOP-4 | 27.23 | 5.700 | **21.12** | 1.113 | 26.72 | 5.469 | **21.89** | 1.008 | 9.00 |
| `Ctrl-Z`† | 26.44 | 5.686 | 20.88 | 0.720 | 26.15 | 5.476 | 21.68 | 0.650 | 2.77 |
| `Ctrl-Z`‡ | **27.34** | **5.705** | 21.02 | **1.138** | **26.73** | **5.501** | 21.84 | **1.025** | 7.72 |
| **Hunyuan-DiT** | | | | | | | | | |
| DDIM | 30.22 | 6.324 | 22.15 | 1.071 | 28.90 | 5.930 | 22.47 | 0.827 | 1.00 |
| Resampling | 30.20 | 6.338 | 22.14 | 1.112 | 28.70 | 5.912 | 22.49 | 0.904 | 2.00 |
| Z-Sampling | 30.10 | **6.398** | 22.18 | 1.111 | 28.60 | **5.997** | 22.49 | 0.899 | 3.00 |
| SOP-1 | 30.56 | 6.333 | 22.19 | 1.232 | 28.77 | 5.902 | 22.50 | 1.102 | 3.00 |
| SOP-4 | 30.81 | 6.303 | 22.22 | **1.444** | 29.49 | 5.942 | 22.58 | 1.246 | 9.00 |
| `Ctrl-Z`† | 30.69 | 6.310 | 22.24 | 1.214 | 29.27 | 5.906 | 22.51 | 1.067 | 2.85 |
| `Ctrl-Z`‡ | **30.91** | 6.325 | **22.30** | 1.441 | **29.79** | 5.908 | **22.62** | **1.319** | 8.79 |

Table 2: Quantitative results of different methods on T2I-CompBench.

| Method | Stable Diffusion 2.1 ↑ | | | | | Hunyuan-DiT ↑ | | | | |
|---|---|---|---|---|---|---|---|---|---|---|
| | Color | Shape | Texture | Spatial | Numeracy | Color | Shape | Texture | Spatial | Numeracy |
| DDIM | 46.27 | 41.01 | 46.06 | 13.80 | 46.44 | 66.46 | 44.32 | 60.06 | 24.18 | 53.99 |
| Resampling | 48.67 | 40.99 | 47.46 | 14.21 | 46.76 | 68.86 | 48.58 | 60.86 | 25.29 | 54.06 |
| Z-Sampling | 50.38 | 42.19 | 48.84 | 13.65 | 47.77 | 68.80 | 48.51 | 60.46 | 22.89 | 54.27 |
| SOP-1 | 53.85 | 45.00 | 51.53 | 18.25 | 50.81 | 71.01 | 49.11 | 63.33 | 21.00 | 54.51 |
| SOP-4 | 59.64 | 48.91 | 60.56 | 16.97 | 52.85 | 73.74 | 53.74 | 64.85 | 21.78 | **57.58** |
| `Ctrl-Z`† | 58.65 | 47.10 | 57.75 | 18.55 | 51.83 | 71.10 | 51.59 | 63.76 | 21.13 | 56.35 |
| `Ctrl-Z`‡ | **61.26** | **53.97** | **62.24** | **19.29** | **53.73** | **73.77** | **54.99** | **66.45** | **25.43** | 56.69 |

## 5.2 QUANTITATIVE EVALUATIONS

The quantitative results in Tabs. 1 and 2 show consistent improvements over baselines brought by `Ctrl-Z` Sampling. In particular, our method achieves significant gains on ImageReward across both Pick-a-Pic and DrawBench. Compared to SOP, which also relies on ImageReward, `Ctrl-Z` Sampling achieves higher HPSv2 and PickScore, indicating stronger improvements in human-aligned image quality. The smaller gain on ImageReward itself is likely due to SOP's always-on search strategy, which repeatedly optimizes the surrogate ImageReward score at each step and thus over-optimizes the reward model without consistently improving overall quality. Tab. 2 shows that `Ctrl-Z` Sampling performs well on CompBench, whose prompts emphasize object relationships and visual attribute binding. `Ctrl-Z` Sampling adaptively increases exploration strength to revisit and escape early-stage, low-frequency local optima in global layout. In contrast, prior methods rely on fixed perturbations and struggle to revise suboptimal trajectories once coarse structures are formed. We leave some further analysis on the result in Appendix Section C.1.

Importantly, `Ctrl-Z` Sampling provides a simple inference-time scaling scheme with a controllable compute–performance trade-off. By tuning its exploration depth and width, it can operate in a low-cost regime (around 3× NFEs of standard DDIM) that outperforms other inference-time guidance baselines, while scaling the budget up to roughly 7–9× NFEs yields further gains, with comparable performance to SOP but fewer computes. Across these NFE settings, `Ctrl-Z` additionally explores along the inversion depth, accessing a richer search context than SOP for comparable or lower over-

Table 3: Ablation results on exploration guided by different reward models and initiation criteria. PS denotes PickScore. Best results are **bolded**; second-best are underlined.

| Method | Pick-a-Pic↑ | | | | DrawBench↑ | | | | NFEs↓ |
|---|---|---|---|---|---|---|---|---|---|
| | HPS v2 | AES | PS | IR | HPS v2 | AES | PS | IR | |
| **Reward Model for Controlled Exploration** | | | | | | | | | |
| CLIPScore | 26.59 | 5.614 | 20.97 | 0.573 | 26.29 | 5.333 | 21.70 | 0.483 | 11.75 |
| HPS v2 | 27.39 | 5.723 | 21.05 | 0.503 | **27.14** | 5.503 | 21.80 | 0.478 | 3.44 |
| AES | 26.29 | **6.147** | 20.87 | 0.306 | 25.54 | **5.902** | 21.62 | 0.231 | 11.17 |
| PickScore | **27.51** | 5.763 | **21.69** | 0.589 | 27.12 | 5.497 | **22.38** | 0.573 | 7.01 |
| ImageReward | 27.34 | 5.705 | 21.02 | **1.138** | 26.73 | 5.501 | 21.84 | **1.025** | 7.72 |
| **Exploration Initiation Criteria** | | | | | | | | | |
| Always ($p = 1.0$) | **27.86** | **5.737** | **21.10** | **1.317** | **27.17** | 5.482 | **21.91** | **1.174** | 16.72 |
| Random ($p = 0.5$) | 27.00 | 5.713 | 21.03 | 1.062 | 26.71 | 5.469 | 21.83 | 0.906 | 7.81 |
| Reward-Based | 27.34 | 5.705 | 21.02 | 1.138 | 26.73 | 5.501 | 21.84 | 1.025 | 7.72 |

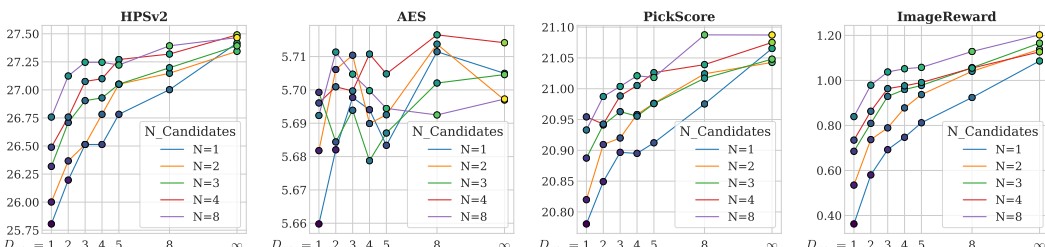

Figure 2: Effect of search depth ($d_{\max}$) and candidates width ($N$) quantitatively. Lighter colors indicate higher NFEs on average. `Ctrl-Z Sampling` benefits from both deeper and wider search, with increased depth sometimes yielding better performance than increased candidates under similar NFEs. We fix $\lambda$ as 30 here for efficiency concerns. The comparisons are best viewed zoomed in.

all computational cost, and thus achieves consistently stronger performance under both small and large compute budgets.

Results on both SD2.1 and Hy-DiT confirm the effectiveness of `Ctrl-Z Sampling` across U-Net and Transformer–based diffusion models, showcasing its compatibility with major frameworks. While our experiments focus primarily on latent diffusion, the hill-climbing intuition naturally extends to pixel-space models such as EDM (Karras et al., 2022), where denoising similarly seeks regions of less noise, higher quality, and better alignment with the input conditions. We expect `Ctrl-Z Sam`-pling to remain effective in such settings, leaving further validation to future work due to constraint on computational resources.

## 5.3 ABLATIONS

The following sections include ablation experiments on the effectiveness of various design and hyper-parameters used in `Ctrl-Z Sampling`. For consistency and efficiency, we set $\delta = 0$ and $\lambda = 40$ for all ablations if not specified. More ablation experiments regarding the max inversion steps, accept threshold, the exploration initiation criteria, and the effect of exploration window can be found at the Appendix Section C.

**Scaling Effect of Exploration Depth and Width.** We investigate how the maximum exploration depth $d_{\max}$ and the number of candidates $N$ per inversion step affect generation quality across evaluation metrics. As shown in Fig. 2, increasing depth allows the model to revisit and revise suboptimal trajectories that commit to structural errors in the early denoising stages, while greater candidate width increase the chance of identifying higher-reward alternatives at each step. These two parameters work in conjunction to enhance the likelihood of generating better-aligned outputs.

We observe that deeper exploration often reduces the reliance on large candidate sets. When $d_{\max}$ is small, increasing $N$ improves performance by providing more local alternatives from different

directions at each step. However, when $d_{\max}$ is large, increasing $N$ does not consistently lead to better results. This suggests that exploration with progressively deeper steps can be more critical for escaping suboptimal trajectories, and that broadening the search with additional candidates provides diminishing returns in such cases. In fact, configurations with greater depth and fewer candidates frequently outperform wider-but-shallow settings under comparable NFEs budgets (as shown by dots with similar colors in Fig. 2), highlighting the importance of adaptively deeper search in enabling efficient and effective inference.

As the exploration parameters increase, consistent improvements are observed across most metrics. An exception is AES, which measures aesthetic quality rather than conditional alignment and is therefore less sensitive to the exploration guided by ImageReward. While large-scale settings are beyond the scope of our study due to resource constraints, the observed trends suggest that `Ctrl-Z` Sampling is also well suited for scalable inference through adaptive exploration.

**Choice of Reward Model.** We evaluate the impact of different reward models using all employed metrics and also CLIPScore (Hessel et al., 2021), with results presented in Tab. 3. It can be observed that `Ctrl-Z` Sampling consistently improves generation quality across all four metrics, regardless of the reward model used. This demonstrates our method's robustness to different guidance objectives.

In particular, using AES as the reward model enhances visual quality. However, since AES does not evaluate condition alignment, its impact on other alignment-sensitive metrics is limited. We observe that PickScore performs better than other CLIP-based models when used to guide generation. Overall, both PickScore and ImageReward lead to stronger performance than other models.

We ultimately adopt ImageReward for the main experiments due to its broader applicability, efficiency, and lighter domain bias, as PickScore is trained on data closely related to the Pick-a-Pic benchmark. While Ma et al. (2024) suggest that combining multiple reward models can further improve performance, it is beyond the scope of our study.

**When to Explore.** This section evaluates different strategies for triggering zigzag exploration. We compare the proposed reward-based initiation mechanism, which activates exploration upon detecting a reward plateau, against two baselines: *Always*, which performs exploration at every denoising step, and *Random*, which triggers exploration at each step with probability $p = 0.5$.

As shown in Tab. 3, the Always strategy yields the highest generation quality but incurs substantial computational overhead, averaging 16.72× NFEs. In contrast, the reward-based approach achieves slightly lower quality with significantly lower cost (7.72× NFEs), offering a more favorable trade-off between quality and efficiency. The Random strategy provides modest quality gains with a similar computational budget (7.81× NFEs), though the probability can be flexibly tuned to interpolate performance between never and always exploring. Overall, the proposed strategy delivers improved sampling quality relative to cost and remains more efficient than the Always strategy. Notably, all three strategies are effective within the `Ctrl-Z` Sampling framework and can be selected based on the desired trade-off between performance and inference cost.

## 5.4 Qualitative Evaluations

Fig. 3 presents qualitative comparisons between baseline methods and the proposed `Ctrl-Z` Sampling. `Ctrl-Z` Sampling produces outputs that are both visually coherent and semantically aligned with input prompts, as reflected in examples involving spatial relations, numeracy, color, and action understanding. In contrast, baseline methods often produce results that are either semantically misaligned with the prompt or lack visual coherence. While SOP can improve image quality by incurring a larger candidate pool, we observe failures in challenging cases. The fixed exploration strength makes SOP vulnerable to low-frequency errors in the latent space, leading to local reward gains from object presence. Consequently, the trajectory often converges to visually plausible but semantically imprecise states (*e.g.*, "cat and vase" or "blue and bear"). In contrast, `Ctrl-Z` Sampling adaptively increases exploration strength, allowing the model to escape such traps and generate more aligned content with distinct low-frequency components. These findings underscore the effectiveness of `Ctrl-Z` Sampling in escaping suboptimal generations through adaptive exploration.

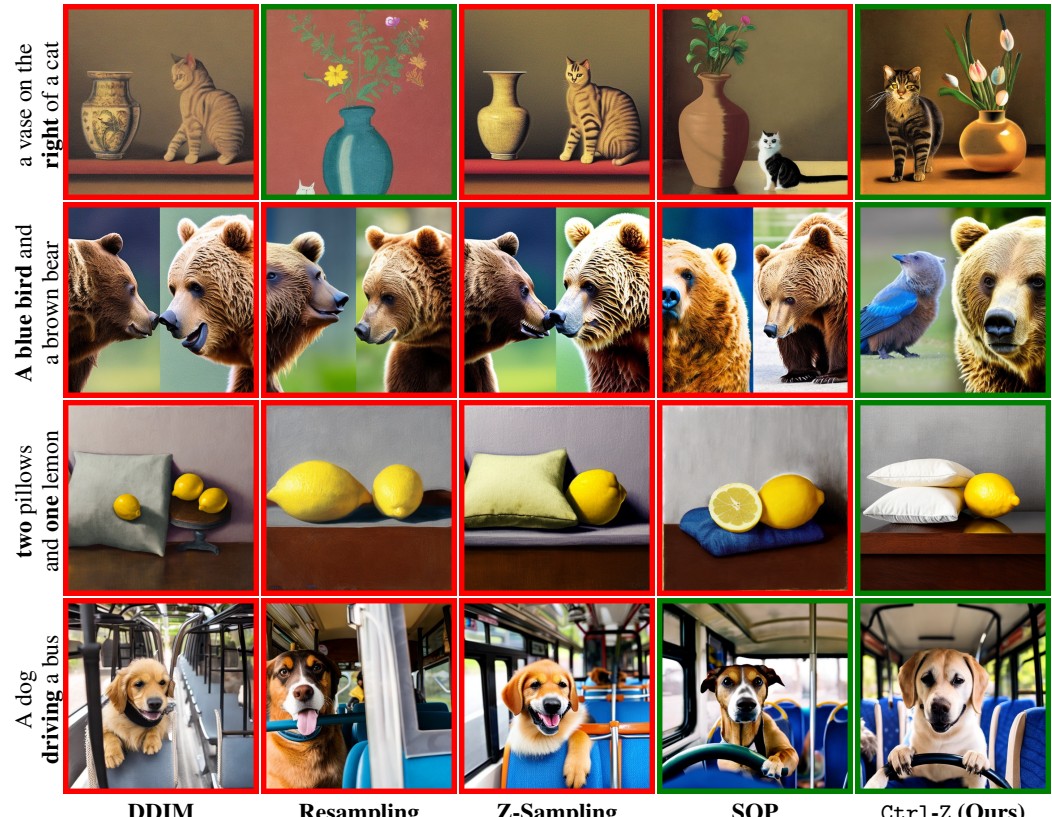

Figure 3: Qualitative comparison of different sampling methods. Generated images that align with the input condition (shown on the left) are highlighted with green bounding boxes, while erroneous or suboptimal images are marked in red.

## 6 CONCLUSION

We present `Ctrl-Z` Sampling, a reward-guided inference-time sampling strategy for diffusion models that improves generation quality by escaping local optima through adaptive noise injection and backward exploration. Experiments on text-to-image benchmarks demonstrate that `Ctrl-Z` Sampling effectively enhances both conditioning alignment and visual fidelity with moderate computational overhead. Future work includes exploring its scalability in broader test-time search settings, and developing unified scheduling strategies that jointly adapt the exploration initiation criteria and strength in a reward-model-agnostic manner.

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

# Appendix

## A  EXTENDED RELATED WORKS

**Efficient Diffusion Sampling.**  Aside from improving generation quality, a complementary line of research focuses on accelerating diffusion sampling. One major approach to this problem is model distillation, where a teacher model is distilled into a faster student (Salimans & Ho, 2022; Katzir et al., 2024; Song et al., 2023). Alternatively, advanced sampling through higher-order ODE integrations and scheduling schemes have been proposed to achieve high-quality outputs with only a few diffusion steps (Karras et al., 2022; Lu et al., 2022; 2025; Zhang & Chen, 2023; Tong et al., 2025). Some works also explore parallel sampling (Shih et al., 2023; Chen et al., 2024b), which improves efficiency by reformulating denoising steps to run concurrently, trading space for time. Inspired by speculative decoding in LLMs, Bortoli et al. (2025) further introduces drafting strategies that efficiently generate multiple candidate trajectories and verify them in parallel. Overall, these methods primarily trade off image quality and memory against inference speed, whereas our method aligns more closely with inference scaling, leveraging additional computations during inference only to improve quality. For this reason, we do not include diffusion accelerators in our comparisons.

**Inference Scaling.**  Rather than allocating additional computation during pretraining (Brown et al., 2020; Kaplan et al., 2020; Hoffmann et al., 2022), recent advances in LLMs have highlighted the potential of inference scaling (or test-time scaling (TTS)) as a paradigm to enhance performance during deployment (Zhang et al., 2025b). Foundational techniques like chain-of-thought prompting encourage step-by-step reasoning to elicit better outputs (Wei et al., 2022), while self-consistency methods generate multiple responses in parallel and aggregate them via voting (Wang et al., 2023). Sequential approaches, such as self-refinement, iteratively revise drafts based on self-feedback (Madaan et al., 2023), and hybrid strategies combine exploration and exploitation, as in ReAct (Yao et al., 2023b), ToT (Yao et al., 2023a), and GoT (Besta et al., 2024). More advanced internal scaling trains models to autonomously extend reasoning (OpenAI, 2024; DeepSeek-AI, 2025), often via bootstrapping (Zelikman et al., 2022) or post-training enhancements (Muennighoff et al., 2025) with algorithms like proximal policy optimization (Schulman et al., 2017).

**Reward-Guided Inference Search.**  Empirical studies reveal scaling laws at inference: increasing test-time compute (*e.g.*, via repeated sampling or deeper search) yields predictable gains akin to pretraining scaling but in a more efficient manner (Brown et al., 2024; Snell et al., 2025). A verifier, *i.e.*, reward model, that scores the quality of intermediate or final outputs is typically used to scale inference by re-ranking or steering candidates. Early frameworks show how verifiers can guide reasoning in LLMs (Cobbe et al., 2021; Creswell et al., 2023). Process reward models (PRMs) enable finer-grained control by scoring intermediate steps (Lightman et al., 2024; Uesato et al., 2022; Setlur et al., 2025; Zhang et al., 2025a), inspiring PRM-guided greedy, beam, BoN, and tree search methods (Zhang et al., 2024; Jiang et al., 2024; Gandhi et al., 2024; Jinnai et al., 2025; Yang et al., 2025). More recent work explores efficiency-oriented strategies such as speculative rejection (Sun et al., 2024) and list-wise re-ranking (Gangi Reddy et al., 2024). Together, these advances demonstrate how inference-time scaling with verifiers can allow smaller models to outperform LLMs with significantly larger parameter sizes (Liu et al., 2025).

While these ideas originated in LLM reasoning, similar inference scaling has been studied for diffusion (Sohl-Dickstein et al., 2015; Ho et al., 2020; Song & Ermon, 2019) and flow models (Lipman et al., 2023; Liu et al., 2023; Tong et al., 2024). Prior efforts optimize denoising trajectories through search (Ma et al., 2025; He et al., 2025; Zhang et al., 2025c) or steer generation with reward-guided objectives (Singhal et al., 2025), improving sample quality, diversity, and alignment at test time without retraining. However, most methods build the candidate pool only through shallow and local perturbations, leaving the search vulnerable to local optima. In contrast, `Ctrl-Z` Sampling adaptively deepens exploration into higher-noise manifolds when stagnation is detected, enabling targeted search where needed and avoiding unnecessary computation. Most existing inference-scaling studies characterize scaling laws under extremely large search budgets. Due to resource constraints, we limit our evaluation to moderate budgets, and leave broader large-scale scaling analyses as an important direction for future work.

## B    EXPERIMENT SETTINGS

### B.1    DATASETS

We evaluate different sampling methods on three text-to-image diffusion benchmarks: **Pick-a-Pic** (Kirstain et al., 2023), **DrawBench** (Saharia et al., 2022), and **T2I-CompBench** (Huang et al., 2023). These benchmarks are designed to assess the performance of diffusion models in generating images from text prompts, each with distinct characteristics to test various aspects of model capabilities.

**Pick-a-Pic** includes 500 diverse, real-user prompts accompanied by human preference annotations. This benchmark emphasizes the diversity of prompts, reflecting a wide range of user-generated inputs and preferences, making it ideal for evaluating how well models handle varied, real-world scenarios.

**DrawBench** comprises 200 curated prompts designed to target compositional and semantic challenges. These prompts focus on complex scenarios, such as combining multiple objects or attributes, testing the models' ability to understand and accurately render intricate textual descriptions.

**T2I-CompBench** (Huang et al., 2023) features 6,000 compositional text prompts, categorized into attribute binding (*e.g.*, color, shape, texture), object relationships (*e.g.*, spatial and non-spatial), and complex compositions. This benchmark is tailored for evaluating open-world compositional text-to-image generation, pushing models to generalize across nuanced combinations of elements. It includes specialized evaluation metrics and explores the use of multimodal large language models (MLLMs) for assessment, providing a comprehensive framework for testing diffusion models' compositional accuracy.

### B.2    METRICS

We evaluate our method using four human-aligned metrics commonly adopted for assessing text-to-image diffusion models.

**HPSv2** (Wu et al., 2023) and **PickScore** (Kirstain et al., 2023) are preference prediction models built on the CLIP (Radford et al., 2021) backbone and fine-tuned on large-scale human preference data. HPSv2 is trained on structured same-prompt image pairs with binary preference labels, enabling robust evaluation of prompt-aligned generation quality. In contrast, PickScore leverages the more diverse and open-ended Pick-a-Pic dataset, which reflects user-generated prompts and preferences collected at scale, thereby improving generalization across prompt styles. Both models outperform the standard CLIPScore (Hessel et al., 2021) in aligning with human judgments.

**ImageReward (IR)** (Xu et al., 2023a) is a learned reward model trained on expert-annotated image–text pairs, designed to produce a scalar score that reflects alignment with human preferences. Unlike HPSv2 and PickScore, which operate on image pairs, ImageReward evaluates each image–prompt pair independently, assigning a continuous reward typically in the range of –2 to +2. This formulation enables both fine-grained assessment and potential use in reinforcement-style optimization.

**Aesthetic Score (AES)** (Schuhmann et al., 2022) estimates the perceptual appeal of images based on large-scale aesthetic annotations. While it provides a lightweight and prompt-agnostic measure of visual quality, AES is not conditioned on the input prompt and thus may fail to reflect semantic alignment or faithfulness to the generation condition.

### B.3    IMPLEMENTATION

We adopt the base version of Stable Diffusion 2.1 (Feng et al., 2023) for U-Net-based and Hunyuan-DiT (Li et al., 2024b) for Transformer-based diffusion models, for their efficiency and effectiveness in text-to-image generation. We set the resolution to $512$ for Stable Diffusion and to $1024$ for Hunyuan-DiT, the number of sampling steps to $T = 50$, All experiments use classifier-free guidance with a scale factor of $5.5$ to ensure stable generation quality. For Ctrl-Z$^{\dagger}$ and Ctrl-Z$^{\ddagger}$, the exploration window is set to $\lambda = 30, 40$, respectively. The maximum number of candidate samples per step is $N = 2, 4$, and the max search depth is set to ($d_{\max} = 2, 3$) to balance quality and efficiency for compute-performance tradeoff under different inference budgets. We use ImageReward (Xu et al., 2023a) as the reward model across all methods, as it has demonstrated strong alignment with human preference in prior evaluations (Ma et al., 2025). The accept threshold is

set to $\delta = 0$. Experiments were conducted on two Linux machines (Ubuntu 22.04). The first used an Intel i7-10700 CPU with an NVIDIA RTX 3090 GPU; the second used an AMD Ryzen 9 7950X3D CPU with an NVIDIA RTX 4090 GPU. Both setups ran with CUDA 12.1. Experiments were implemented in Python using PyTorch (v2.4.0) and Diffusers (v0.32.2).

### B.4 BASELINES

We compare `Ctrl-Z` Sampling with several existing sampling strategies, including the standard **DDIM** sampling (Song et al., 2021), **Resampling** (Lugmayr et al., 2022), **Z Sampling** (Bai et al., 2024), **Search over Paths (SOP)** (Ma et al., 2025), as introduced in the main paper (Fig. 1). The details on these methods are as follows:

**Resampling** (Lugmayr et al., 2022) perturbs the current intermediate representation by reintroducing noise, hence stepping backward in the diffusion process to reach a higher-noise state. From this noisier point, the model continues denoising again, which allows it to better incorporate conditioning information and potentially generate outputs that are more aligned with the conditional input.

**Z Sampling** (Bai et al., 2024) perturbs the current latent by stepping backward to a higher-noise state. At each step, it estimates the forward generation trajectory under weak or no classifier-free guidance, which typically leads to visually plausible but less aligned outputs. It then injects noise in the opposite direction of this estimated denoising path, effectively inverting the model's default tendency toward local optima. This targeted reversal helps escape alignment traps and improves consistency with the conditional input during subsequent denoising.

**Search Over Path (SOP)** (Kim et al., 2025) reverses the current latent to a higher-noise state, where it samples multiple perturbed candidates by injecting different noise patterns at each step. Each candidate is then denoised forward, and a reward model evaluates their alignment with the conditional input. The best-performing path is selected to continue generation. This strategy enables SOP to improve sampling by exploring multiple nearby directions and explicitly selecting the most promising one based on reward. The number of search candidates in SOP per step is set to 4 to ensure an overall comparable NFEs to the default settings of `Ctrl-Z` Sampling.

Various other sampling techniques such as (Zhao et al., 2023; Sabour et al., 2024) are not included for comparison as they are considered orthogonal and may be combined with the inversion-driven exploration methods discussed here. In Section C.5, we provide studies on how combining other orthogonal sampling techniques with `Ctrl-Z` Sampling will affect performance.

## C ADDITIONAL EXPERIMENTS AND RESULTS

### C.1 FURTHER ANALYSIS ON THE QUANTITATIVE RESULTS

While the quantitative evaluation in Tab. 1 primarily demonstrates the effectiveness of `Ctrl-Z` Sampling in guided exploration, several additional observations are noteworthy. With Stable Diffusion 2.1, `Ctrl-Z` Sampling guided by ImageReward yields substantial improvements not only in alignment metrics but also in AES. This can be attributed to the weaker baseline quality of SD2.1, where enhancements in human preference alignment are often accompanied by perceptual gains. In contrast, Hunyuan-DiT, with its larger capacity and higher-resolution outputs, already delivers strong aesthetic quality. In this setting, ImageReward, which is trained to capture human preference, primarily emphasizes improvements in alignment rather than aesthetics when comparing candidate states. As a result, applying `Ctrl-Z` Sampling to Hunyuan-DiT consistently improves IR, the guiding metric, but may lead to a slight reduction in AES.

It is also worth noting that, under identical search parameters, `Ctrl-Z` Sampling with Hunyuan-DiT requires about one additional forward call per sampling step on average (8.79 NFEs vs. 7.72 for Stable Diffusion 2.1). A plausible explanation is that Hunyuan-DiT, being a larger and more capable model, produces higher-quality denoising outputs at each step, making it harder for additional exploration to exceed the reward threshold (i.e., $R([c], \hat{x}_0^{t-1}) \geq r_{\text{prev}} + \delta$). At first glance, this may seem counterintuitive: a stronger model might be expected to achieve larger reward improvements and thus initiate exploration less often. However, the observed gains in ImageReward for Hunyuan-DiT are smaller than those for SD2.1, indicating that its denoising outputs are already closer to optimal. Consequently, finding further improvements through exploration is more challenging, which results in slightly stronger adaptive explorations with larger inversion step size and also more computation.

While `Ctrl-Z` Sampling still outperforms SOP in this setting, the variability in sampling cost introduces a potential weakness compared to SOP, whose cost is fixed. Nonetheless, this issue can be mitigated through strategies such as adaptive pruning or early termination of deeper searches, helping to stabilize inference cost across models of different capacities.

An additional observation from Tab. 2 is that `Ctrl-Z` Sampling outperforms SOP at most cases except for numeracy prompts on Hunyuan-DiT, where SOP outperforms `Ctrl-Z` Sampling, likely because its fixed local perturbations are sufficient to recover enumerations when small object instances are concentrated in high-resolution outputs. Nonetheless, the proposed dynamic adaptive exploration scheme also incorporates local optimizations, yielding numeracy results comparable to SOP while substantially improving performance on other subsets. Conversely, SOP performs poorly on spatial prompts. Interestingly, Resampling, which also injects noise more randomly but without reward-based guidance, demonstrates notable improvements in spatial-related prompts. This suggests a potential limitation of SOP: its reliance on reward signals at every step may contribute to early reward saturation, where strong reward gains from object presence encourages the model to commit prematurely to locally optimal states. Although these states align well with the reward model, they may fall short in capturing the broader spatial relationships intended by the prompt. In comparison, the proposed `Ctrl-Z` Sampling method applies reward-guided exploration adaptively with less frequency and increasing strength. The selective exploration reduces early saturation, while the ability to increase exploration strength helps escape locally optimal states and recovers more coherent spatial structures.

## C.2 MAX INVERSION STEPS AND ACCEPT THRESHOLD

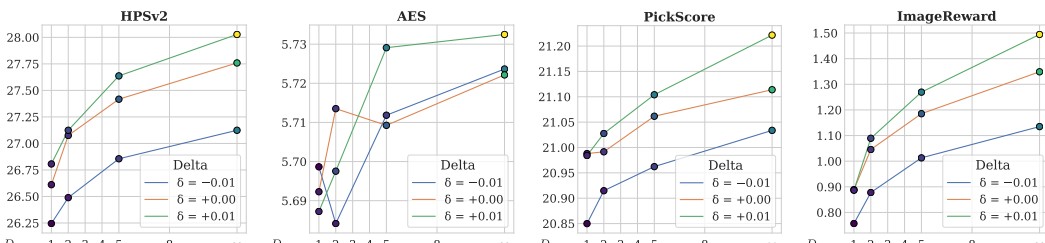

Figure 4: Evaluation metrics across different accept threshold $\delta = -1, 0, 1$ for different max search depth $d_{max} \in \{1, 2, 5, \infty\}$.

This section examines the impact of the acceptance threshold ($\delta$) and maximum inversion depth ($d_{\max}$), with results shown in Fig. 4. In our design, $\delta$ controls both when to initiate exploration and whether to accept each inversion step, requiring the reward to exceed that of the previous step by at least $\delta$ (which may be negative to allow slight decreases).

We find that increasing either parameter generally improves generation quality, supporting the effectiveness and scalability of deeper, more permissive zigzag exploration. These improvements align with our adaptive control design for flexible inference-time scaling. Aesthetic Score (AES), however, shows less consistent gains, likely due to its emphasis on visual appeal over conditional alignment. Notably, when $\delta = 0.01$, which enforces stricter acceptance, AES improves more reliably with increased $d_{\max}$. This suggests that stricter thresholds filter out superficial reward increases (such as those resulting from overfitting to prompt-specific details) and instead favor candidate steps that introduce more substantive changes to the generation trajectory. These structural improvements may contribute to better global coherence and visual quality, which are more aligned with what AES measures.

Overall, $\delta$ provides a flexible mechanism for balancing quality and cost. However, requiring a fixed stepwise reward improvement can introduce instability, especially when early steps yield high rewards, making future acceptance more difficult and potentially causing unnecessary explorations with negligible benefit. Future work could explore global reward scheduling to adaptively stabilize this process.

Table 4: Ablation results on images generated with different exploration initiation criteria under similar NFEs settings. The Reward-based strategy that initiates exploration on detected plateau demonstrates comparable performance to the ones that 'Always' initiate exploration at each step, with overall fewer NFEs.

| Parameters | Pick-a-Pic ↑ | | | | DrawBench ↑ | | | | NFEs ↓ |
|---|---|---|---|---|---|---|---|---|---|
| $(d_{max}, N, \lambda)$ | HPS v2 | AES | PickScore | IR | HPS v2 | AES | PickScore | IR | |
| **Always Explore** | | | | | | | | | |
| (3, 2, 40) | 27.44 | 5.691 | **21.09** | 1.189 | **26.98** | 5.436 | 21.85 | **1.076** | 10.00 |
| (2, 3, 40) | **27.54** | 5.683 | 21.07 | **1.199** | 27.10 | 5.459 | **21.93** | 1.049 | 9.46 |
| (1, 4, 40) | 27.05 | 5.689 | 21.00 | 1.057 | 26.59 | 5.484 | 21.78 | 0.813 | 7.24 |
| (3, 3, 30) | 27.34 | 5.704 | 21.04 | 1.132 | 26.98 | 5.469 | 21.86 | 0.949 | 9.90 |
| **Reward-based** (Ours) | | | | | | | | | |
| (3, 4, 40) | 27.34 | **5.705** | 21.02 | 1.138 | 26.73 | **5.501** | 21.84 | 1.025 | 7.72 |

Table 5: Ablation results for randomly initiating the exploration process with varying probabilities. $p = 0$ corresponds to standard DDIM sampling, while $p = 1$ is equivalent to the 'Always' exploration strategy. Increasing the probability consistently improves generation quality, accompanied by a corresponding increase in function evaluations (NFEs).

| Exploration | Pick-a-Pic ↑ | | | | DrawBench ↑ | | | | NFEs ↓ |
|---|---|---|---|---|---|---|---|---|---|
| **Probability** | HPS v2 | AES | PickScore | IR | HPS v2 | AES | PickScore | IR | |
| $p = 0$ | 25.34 | 5.649 | 20.67 | 0.194 | 24.90 | 5.410 | 21.39 | 0.046 | 1.00 |
| $p = 0.1$ | 25.83 | 5.677 | 20.78 | 0.474 | 25.29 | 5.445 | 21.52 | 0.347 | 2.12 |
| $p = 0.3$ | 26.54 | 5.705 | 20.89 | 0.801 | 26.37 | 5.470 | 21.70 | 0.758 | 4.68 |
| $p = 0.5$ | 27.00 | 5.713 | 21.03 | 1.062 | 26.71 | 5.469 | 21.83 | 0.906 | 7.81 |
| $p = 0.7$ | 27.37 | 5.695 | 21.04 | 1.193 | 26.90 | 5.471 | 21.83 | 1.038 | 11.11 |
| $p = 1$ | **27.86** | **5.737** | **21.10** | **1.317** | **27.17** | **5.482** | **21.91** | **1.174** | 16.72 |

## C.3 Exploration Initiation Criteria

**Always Initiate Exploration.** To more thoroughly evaluate the effectiveness of the proposed reward-based exploration initiation criterion, we conduct a set of comparative experiments using an 'Always' explore strategy, where exploration is triggered at every diffusion step. To ensure a fair comparison, we reduce the associated exploration parameters. Specifically, we reduce the maximum exploration depth $d_{\max}$, the number of candidate paths $N$, or exploration window $\lambda$, to achieve a similar average number of function evaluations (NFEs) as those under the proposed settings with SD2.1. The results are presented in Tab. 4.

Despite the reduced exploration budget, most configurations under the 'Always' strategy still result in higher NFEs on average. While they offer slightly improved prompt alignment, our proposed method consistently achieves better aesthetic scores. This suggests that always triggering exploration may lead to excessive optimization of reward values, potentially overfit to prompt-specific or reward-specific features and degrading visual quality. In contrast, the reward-based initiation strategy selectively activates exploration, striking a better balance between conditional alignment and perceptual quality. This indicates its robustness in avoiding over-optimization and improving aesthetic appeal.

**Randomly Initiate Exploration.** Moreover, we also study the effect of randomly initiate the exploration process with different probabilities, the results are demonstrated in Tab. 5. As the probability increases, generation quality improves across all metrics, including aesthetic score (AES), with $p = 1.0$ achieving the best performance. However, this also leads to substantially higher NFEs.

In contrast to our budget-controlled comparison with the 'Always' strategy, the random initiation setting allows NFEs to scale freely. As such, the observed gains at higher $p$ are largely attributable to increased computational budget rather than the effectiveness of the initiation strategy itself. While

Table 6: Quantitative results under different $\lambda$ values for ImageReward-guided controlled exploration. NFE indicates the average number of function evaluations. Overall, generation quality improves as the $\lambda$ value increases.

| $\lambda$ | Pick-a-Pic↑ | | | | DrawBench↑ | | | | NFEs ↓ |
| | HPS v2 | AES | PickScore | IR | HPS v2 | AES | PickScore | IR | |
|---|---|---|---|---|---|---|---|---|---|
| 10 / 50 | 26.05 | 5.669 | 20.81 | 0.478 | 25.70 | 5.415 | 21.52 | 0.374 | 2.21 |
| 20 / 50 | 26.64 | 5.693 | 20.93 | 0.748 | 26.20 | 5.442 | 21.67 | 0.669 | 3.83 |
| 30 / 50 | 27.08 | 5.700 | 20.99 | 0.964 | 26.60 | 5.435 | 21.78 | 0.874 | 5.67 |
| 40 / 50 | 27.34 | 5.705 | 21.02 | 1.138 | 26.73 | 5.501 | 21.84 | 1.025 | 7.72 |
| 50 / 50 | 27.29 | 5.714 | 21.03 | 1.286 | 26.80 | 5.468 | 21.80 | 1.170 | 10.19 |

less efficient than our reward-based method, random initiation can serve as a flexible alternative when computational resources are not a primary constraint.

### C.4 EFFECT OF EXPLORATION WINDOW $\lambda$

Table 6 reports the effect of varying the exploration window $\lambda$. As shown, increasing $\lambda$ generally improves performance, particularly in terms of the reward model metric (IR), which directly guides exploration. However, the performance gains gradually diminish as $\lambda$ becomes large, with some metrics even exhibiting slight degradation. Notably, the number of additional NFEs also increases substantially under larger $\lambda$ values.

We attribute this behavior to the tendency of late-stage explorations to 'overfit' to the reward model. When the latent representation has already committed to local optima, `Ctrl-Z` Sampling exhaustively searches for marginal reward improvements, which leads to increased computational cost and potential performance drop on metrics not directly optimized by the reward. This resembles classical overfitting, where excessive model flexibility results in reduced generalization and efficiency. On the other hand, limiting exploration to the early stages, with smaller $\lambda$ values, can be viewed as initiating exploration before the trajectory descends too far into a particular basin of the loss landscape. At this point, the generation remains relatively flexible and is less constrained by local optima. As a result, `Ctrl-Z` Sampling is more likely to discover alternative directions that lead toward better optima, enabling more efficient and effective convergence. Based on this trade-off, we adopt $\lambda = 40$ as a default, which achieves decent aesthetic quality besides condition alignment, as well as a favorable balance between performance and computational overhead.

### C.5 COMBINING `Ctrl-Z` SAMPLING WITH ADDITIONAL OPTIMIZATION METHODS

`Ctrl-Z` Sampling enhances diffusion sampling by inverting to earlier timestep with random perturbations and selecting promising candidates using reward-based metrics. It is complementary to sampling methods that optimize the forward process or scheduling, and can be seamlessly combined with them. In this section, we study its integration with two representative approaches: AYS sampling (Sabour et al., 2024), which improves scheduling, and CFG++ (Chung et al., 2024), which refines classifier-free guidance with manifold constraints. We do not examine forward-process variations here, as experimenting with different diffusion architectures already constitutes a form of forward improvement.

#### C.5.1 `Ctrl-Z` SAMPLING + DPM/AYS SAMPLING

DPM-Solver (Lu et al., 2022) accelerates diffusion sampling by exploiting the semi-linear structure of the probability flow ODE. Instead of discretizing both linear and nonlinear terms as in generic solvers, it solves the linear component analytically and applies exponential integrator techniques to approximate the nonlinear term. The first-order variant (similar to DDIM) uses a single noise prediction per step, while the practical second-order solver introduces an intermediate state to refine the estimate and substantially reduce discretization error.

AYS (Align Your Steps) (Sabour et al., 2024) is a method for optimizing diffusion sampling schedules. Instead of relying on hand-crafted heuristics like cosine or polynomial timesteps, it formulates

Table 7: Quantitative evaluation results when combining `Ctrl-Z Sampling` with AYS sampling on Pick-a-Pic and DrawBench.

| Method | Pick-a-Pic↑ | | | | DrawBench↑ | | | |
|---|---|---|---|---|---|---|---|---|
| | HPS v2 | AES | PickScore | IR | HPS v2 | AES | PickScore | IR |
| DPM | 23.08 | 5.478 | 20.20 | -0.203 | 22.92 | 5.190 | 20.888 | -0.312 |
| AYS | 23.46 | 5.460 | 20.26 | -0.145 | 23.34 | 5.190 | 20.942 | -0.191 |
| Ctrl-Z + DPM | 24.23 | **5.494** | 20.39 | 0.158 | **24.17** | **5.240** | **21.104** | 0.088 |
| Ctrl-Z + AYS | **24.94** | 5.450 | **20.42** | **0.433** | 24.06 | 5.170 | 20.995 | **0.146** |

Table 8: Quantitative evaluation results when combining `Ctrl-Z Sampling` with AYS sampling on T2I-CompBench.

| Method | Color ↑ | Shape ↑ | Texture ↑ | Spatial ↑ | Numeracy ↑ |
|---|---|---|---|---|---|
| DPM | 35.98 | 36.53 | 42.41 | 9.26 | 43.50 |
| AYS | 35.96 | 36.71 | 42.66 | 8.87 | 42.98 |
| Ctrl-Z + DPM | 37.78 | 37.87 | 43.14 | **10.70** | 44.47 |
| Ctrl-Z + AYS | **39.02** | **39.73** | **46.68** | 10.24 | **45.47** |

schedule selection as an optimization problem using stochastic calculus, yielding solver-specific schedules that significantly improve output quality, especially in few-step regimes.

**Implementation.** We adopt the AYS schedule based on the 10-step second-order DPM solver (Lu et al., 2022), as it is the default scheduler for AYS and the only one publicly available. At inference, AYS is implemented by replacing the default diffusion timesteps with the optimized schedule. Since the AYS schedule is model-specific, we use Stable Diffusion 1.5 in this section to align with the released parameters. For `Ctrl-Z Sampling`, we set the inversion strength to match the schedules in DPM or AYS and apply identical formulations for $x_0$ prediction and other steps as described in the main text.

With only 10 sampling steps, each step yields substantial reward gains toward the final image, making it difficult to design an effective exploration threshold $\delta$. Under the default criterion ($\delta = 0$), exploration would rarely be triggered, as subsequent steps almost always improve reward. Therefore, in these experiments we initiate exploration at every step.

**Results and Discussion.** Quantitative results on Pick-a-Pic, DrawBench, and CompBench for combining `Ctrl-Z Sampling` with DPM and AYS are reported in Tabs. 7 and 8. By replacing the default scheduler with learned timesteps, AYS yields substantial improvements over DPM on Pick-a-Pic and DrawBench. However, its advantage is less evident on CompBench, where DPM achieves stronger performance on the Spatial and Numeracy subsets. These trends remain largely consistent when either scheduler is combined with `Ctrl-Z Sampling` and showing improved performance, supporting our expectation that `Ctrl-Z Sampling` enhances diffusion sampling by escaping local optima and remains orthogonal to the choice of scheduling strategy.

While `Ctrl-Z Sampling` improves generation quality on both DPM and AYS, the gains are smaller than with standard 50-step DDIM sampling, particularly on CompBench. This is largely due to the coarse step size in the DPM solver: each step removes substantial noise and already yields strong reward gains. As a result, even when exploration is initiated, deeper adaptive exploration is seldom triggered, since higher-reward alternatives are often accessible with only one backward step. In contrast, with more sampling steps, as in DDIM, some denoising updates differ only marginally in direction, making exploration more effective by forcing the model to search harder for alternative states and thereby escape local optima. On CompBench, where capturing structure-relevant low-frequency information is critical, this explains why `Ctrl-Z Sampling` contributes less when paired with large-step solvers. Although raising the acceptance threshold $\delta$ could partially alleviate the issue, tuning it under coarse schedules remains challenging. Overall, these factors account for the diminished gains when combining `Ctrl-Z Sampling` with 10-step AYS and highlight the need for more flexible scheduling of exploration initiation and acceptance criteria.

Table 9: Quantitative evaluation results when combining `Ctrl-Z` Sampling with CFG++ on Pick-a-Pic and DrawBench. `Ctrl-Z` ↑ indicates the `Ctrl-Z` experimented with larger exploration parameters.

| | Pick-a-Pic↑ | | | | DrawBench↑ | | | |
|---|---|---|---|---|---|---|---|---|
| **Method (DDIM)** | HPS v2 | AES | PickScore | IR | HPS v2 | AES | PickScore | IR |
| + CFG | 25.34 | 5.649 | 20.67 | 0.194 | 24.90 | 5.410 | 21.39 | 0.046 |
| + CFG++ | 25.78 | 5.675 | 20.72 | 0.276 | 25.39 | 5.452 | 21.46 | 0.189 |
| + `Ctrl-Z` + CFG | 27.34 | **5.705** | 21.02 | 1.138 | 26.73 | **5.501** | **21.84** | 1.025 |
| + `Ctrl-Z` $+_a$ CFG++ | 26.68 | 5.675 | 20.88 | 0.968 | 26.00 | 5.470 | 21.63 | 0.753 |
| + `Ctrl-Z` $+_p$ CFG++ | **27.44** | 5.696 | **21.02** | **1.217** | **26.86** | 5.434 | 21.81 | **1.101** |

Table 10: Quantitative evaluation results when combining `Ctrl-Z` Sampling with CFG++ on T2I-CompBench.

| **Method (DDIM)** | Color ↑ | Shape ↑ | Texture ↑ | Spatial ↑ | Numeracy ↑ |
|---|---|---|---|---|---|
| +CFG | 46.27 | 41.01 | 46.06 | 13.80 | 46.44 |
| +CFG++ | 49.25 | 42.89 | 46.65 | 14.00 | 45.75 |
| + Ctrl-Z + CFG | 61.26 | 53.97 | 62.24 | **19.29** | **53.73** |
| + Ctrl-Z $+_a$ CFG++ | 58.89 | 47.85 | 57.46 | 16.93 | 51.03 |
| + Ctrl-Z $+_p$ CFG++ | **62.76** | **54.04** | **62.46** | 19.25 | 53.65 |

### C.5.2 `Ctrl-Z` SAMPLING + CFG++

A known limitation of standard classifier-free guidance (CFG) is that the guided prediction $\epsilon_\theta^t(x_t, c)$ extrapolates away from the unconditional branch. More explicitly, the two outputs can be combined through a guidance weight $\omega$ as:

$$\epsilon_\theta^{t,\omega}(x_t) = \epsilon_\theta^t(x_t, \varnothing) + \omega\big(\epsilon_\theta^t(x_t, c) - \epsilon_\theta^t(x_t, \varnothing)\big), \tag{6}$$

and for $\omega > 1$ this extrapolation tends to push the trajectory off the diffusion manifold, resulting in instability and poor inversion. While DDIM with CFG applies the guided noise as follows,

$$\hat{x}_0 = \frac{x_t - \sqrt{1 - \bar{\alpha}_t}\,\epsilon_\theta^t(x_t, c)}{\sqrt{\bar{\alpha}_t}}, \quad x_{t-1}^{\text{CFG}} = \sqrt{\bar{\alpha}_{t-1}}\,\hat{x}_0 + \sqrt{1 - \bar{\alpha}_{t-1}}\,\epsilon_\theta^t(x_t, c), \tag{7}$$

CFG++ (Chung et al., 2024) makes a simple but effective change: the clean estimate $\hat{x}_0$ still leverages the guided noise, but the re-noising step reverts to the unconditional prediction:

$$x_{t-1}^{\text{CFG++}} = \sqrt{\bar{\alpha}_{t-1}}\,\hat{x}_0 + \sqrt{1 - \bar{\alpha}_{t-1}}\,\epsilon_\theta^t(x_t, \varnothing). \tag{8}$$

Intuitively, this replaces *extrapolation* with *interpolation* between the conditional and unconditional paths, keeping the trajectory closer to the data manifold. The result is more stable sampling and easier inversion, making CFG++ a natural companion to additional guidance mechanisms.

**Implementation.** We evaluate two variants. In `Ctrl-Z` $+_{a(ll)}$ CFG++, we replace CFG with CFG++ in all forward computations of `Ctrl-Z` Sampling (*i.e.*, Lines 6 and 19 in Alg. 1). In `Ctrl-Z` $+_{p(artial)}$ CFG++, we apply CFG++ only to the standard denoising steps (Line 6), while retaining traditional CFG for the exploration steps. We use $\omega = 0.6$ as in the official CFG++ implementation. Experiments are conducted on all benchmarks with all other parameters identical to the main text.

**Results and Discussion.** The quantitative results are presented in Tabs. 9 and 10. While CFG++ consistently outperforms traditional CFG across benchmarks, combining `Ctrl-Z` Sampling with CFG++ on *all* denoising steps yields smaller gains than when paired with CFG. This outcome can be explained by two factors. First, CFG++ smooths the denoising trajectory and mitigates off-manifold deviations, which reduces the number of sharp local optima that `Ctrl-Z` Sampling is designed to exploit through inversion and adaptive exploration (effectively constraining exploration by 'chaining' the diffusion process to the unconditional denoising direction). Second, CFG++ has its strongest

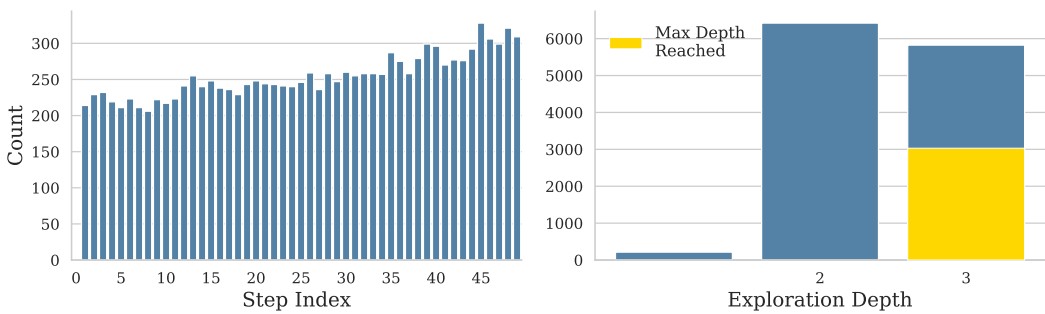

Figure 5: Left: Count of explorations initiated at each diffusion step. Right: Count of final exploration depths when adaptive exploration terminates; the yellow bar indicates cases where termination occurs due to reaching the maximum depth limit.

influence in the early denoising stages where `Ctrl-Z` Sampling is also most effective, as structural errors at this stage propagate through the entire trajectory. By stabilizing these early steps, CFG++ preemptively limits the opportunities for `Ctrl-Z` Sampling to discover alternative high-reward trajectories. Together, these factors account for the diminished marginal benefit observed when the two methods are combined.

These deductions are further supported by experiments where `Ctrl-Z` Sampling is combined with CFG++ only *partially*, leaving the exploration denoising steps unmodified. In this setting, the aforementioned limitations are alleviated, as `Ctrl-Z` Sampling can still explore off-manifold regions to identify promising alternative trajectories. This hybrid strategy leverages the strengths of both methods: in-manifold updates from CFG++ are used when reward signals steadily improve, while `Ctrl-Z` Sampling activates off-manifold explorations to escape local optima when progress stalls. Consequently, `Ctrl-Z` $+_p$ CFG++ achieves stronger overall performance than its counterparts.

### C.6 DISTRIBUTION OF EXPLORATION STEPS.

To analyze the behavior of `Ctrl-Z` Sampling, we record the frequency of exploration initiations across sampling steps and the final inversion depth reached when adaptive exploration terminates. Results are accumulated over 500 prompts with $\lambda = 50$, $N = 4$, and $d_{\max} = 3$, as shown in Fig. 5. The left plot shows that explorations, triggered by the criterion $R(c, \hat{x}_0^{t-1}) < r_{\text{prev}} + \delta$ (Equation (5)), are distributed throughout the trajectory but occur more frequently at later steps. We attribute this to reward saturation: as $\hat{x}_0^{t-1}$ changes only marginally with larger timesteps, rewards plateau and explorations are triggered more often. This observation highlights the benefit of adopting a smaller $\lambda$, which improves efficiency by avoiding unnecessary late-stage explorations with limited effect. For detailed analyses and visualizations of how explorations impact generation quality at different steps, see Section F.2.

From the right plot, we observe that with four exploration candidates, the process rarely terminates after a single inversion step; more often, `Ctrl-Z` Sampling identifies better candidate states with an inversion depth of two. A substantial number of explorations also terminate at the maximum depth of three, often because no more satisfying state is found, yet the search is upper-bounded at the specific depth and terminated (as indicated by the yellow bar). Since computational cost grows linearly with depth, we cap inversion at three steps. Overall, the uneven distribution of exploration depths highlights the importance of adaptive depth, which allocates compute and exploration strength flexibly according to the difficulty of each case.

## D LIMITATIONS AND FUTURE WORKS

**Global Scheduling of Exploration Criteria.** A key limitation of the current framework lies in the use of a fixed per-step acceptance threshold $\delta$ to control both the initiation and progression of zigzag exploration. This design assumes that a meaningful improvement in reward is always possible at each step, but in practice, diffusion trajectories often pass through intermediate states that are already

near-optimal under the given reward model. In such cases, requiring a fixed threshold improvement may be overly strict. As a result, exploration may overfit to intermediate states that align well with the specific reward model but fail to yield tangible improvements in overall generation quality. This limitation can be further exacerbated when a large reward improvement is required too early in the generation process. In such cases, the model may pass through intermediate states that already satisfy prominent reward features, such as object presence or visual plausibility, making further improvements in global structure or semantic alignment difficult to achieve.

Future work could study globally adaptive exploration strategies that adjust explore initiations and depth. For example, the system could track reward gradients, or normalized improvements to determine whether stricter or more permissive criteria should be applied at a given step. This would help the model recognize when it has entered a plateau or near-optimal region, and respond appropriately, for example by relaxing acceptance conditions or terminating exploration more effectively. In addition, we may consider reward scheduling mechanisms that explicitly guide the expected reward progression over time. These could enforce soft constraints, such as targeting specific reward levels at certain diffusion steps. Ideally, these mechanisms would remain general enough to apply across different reward models and diffusion pipelines.

However, in this work, we focus on effective exploration strategies and intentionally adopt a simple scheduling formulation to demonstrate that even basic adaptive criteria already yield substantial benefits. This provides a solid foundation for future extensions incorporating more sophisticated scheduling for diffusion sampling with controlled explorations.

**Experimental Validations on Inference Scaling.** While we demonstrate the effectiveness of adaptive exploration under moderate computational budgets, current evaluation does not fully capture performance at larger scales due to resource constraints. Future work could investigate broader configurations, including deeper exploration depths, increased candidate pool sizes, and stricter acceptance thresholds, to better understand how the proposed method scales with inference cost. A more comprehensive evaluation may reveal whether the quality-efficiency trade-offs follow predictable trends, as suggested by recent work on test-time scaling in diffusion models. These insights would guide the design of inference-time strategies adaptable to different budget constraints.

## E  LLM USAGE STATEMENT

We used GPT-4.5 and GPT-5 exclusively during the writing phase of this paper, for the purpose of polishing the manuscript. Specifically, we provided prompts such as: "Revise the grammar and academic wording of this paragraph, and list aspects to be improved, including suggestions on how to improve them." All outputs were subsequently reviewed and manually revised to ensure that technical accuracy was preserved. No part of the research ideation or experimental analysis relied on LLM assistance.

## F  QUALITATIVE ANALYSIS

### F.1  MORE QUALITATIVE SAMPLES

Figure 6 presents a set of additional qualitative results generated using different sampling methods. As shown, the proposed `Ctrl-Z` Sampling consistently produces the most visually coherent and prompt-aligned outputs across a variety of test cases. In contrast, the baseline methods often fail to maintain global consistency, producing outputs that may appear locally plausible but deviate from the intended semantics of the prompt. These results highlight the effectiveness of `Ctrl-Z` Sampling in generating globally aligned images, which we attribute to its adaptive mechanism for escaping local optima. This mechanism enables the method to avoid premature convergence to suboptimal visual representations, thereby mitigating the common failure mode of generating outputs that are visually appealing but misaligned with the conditional input.

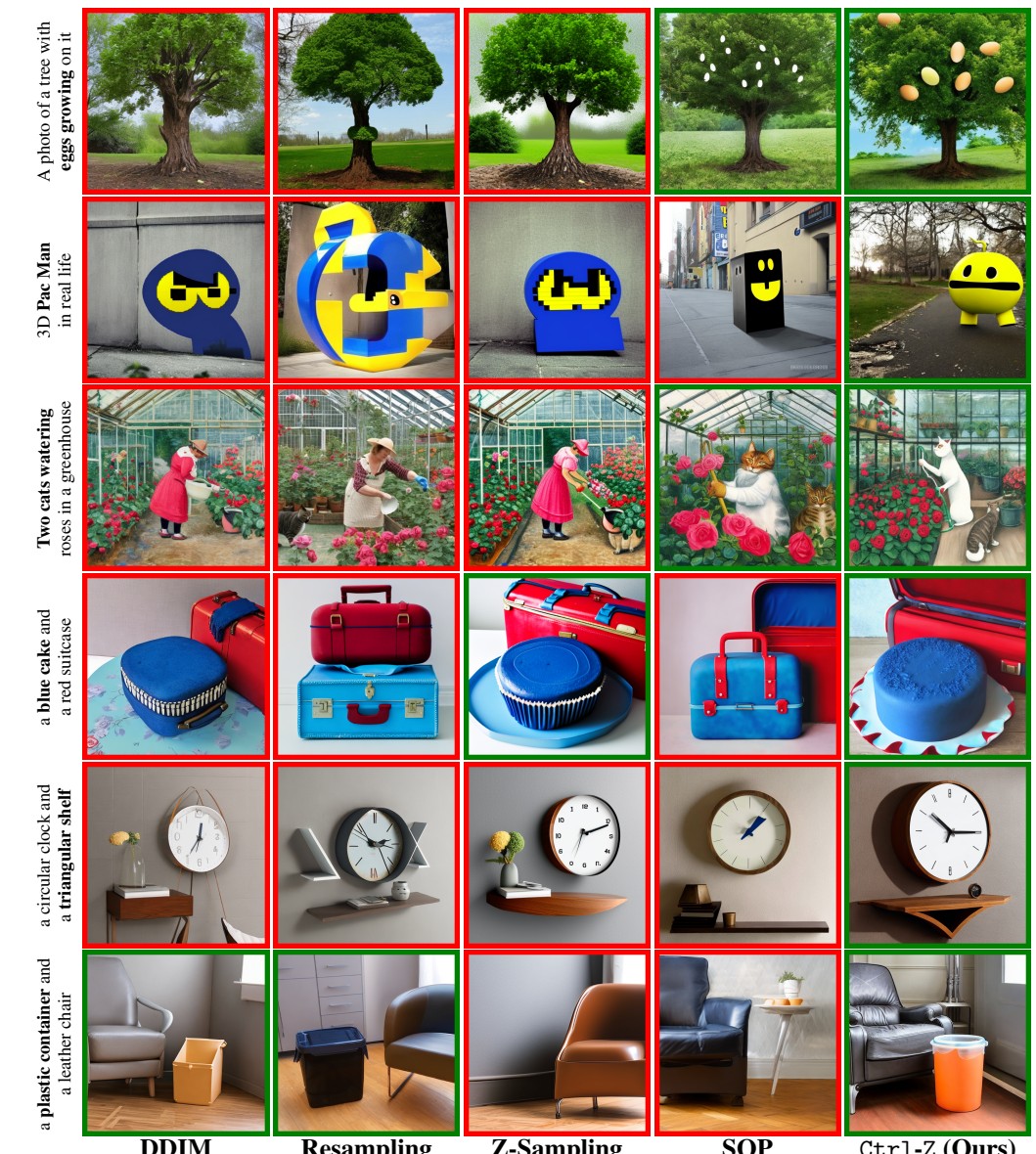

Figure 6: Qualitative comparison of different sampling methods. Our proposed `Ctrl-Z` Sampling generates more coherent and condition-aligned outputs.

## F.2 SAMPLES ON INTERMEDIATE EXPLORATION STEPS

We first present in Fig. 7 a tree-structured visualization of the sampling path during the first 10 steps, comparing our `Ctrl-Z` Sampling with standard DDIM. While DDIM follows a single greedy optimization path determined by the diffusion prior, `Ctrl-Z` Sampling performs controlled zigzag explorations that branch out when the reward fails to improve. This enables the model to escape local optima states and converge to a high-reward state that satisfies the conditional alignment. In all visualizations, steps where the stepwise reward shows insufficient improvement compared to the previous step and exploration is initialized (i.e., $R([c], \hat{x}_0^{t-1}) < r_{\text{prev}} + \delta$ with $\delta = 0$) are highlighted in red, while steps without explorations are shown in green.

We then provide more detailed visualizations of the predicted $\hat{x}_0^{t-1}$ and their associated rewards in Figs. 8 to 11. For each step, we show the decoded intermediate prediction together with its step index and reward. Only the first 30 out of 50 total sampling steps are displayed, as the diffusion

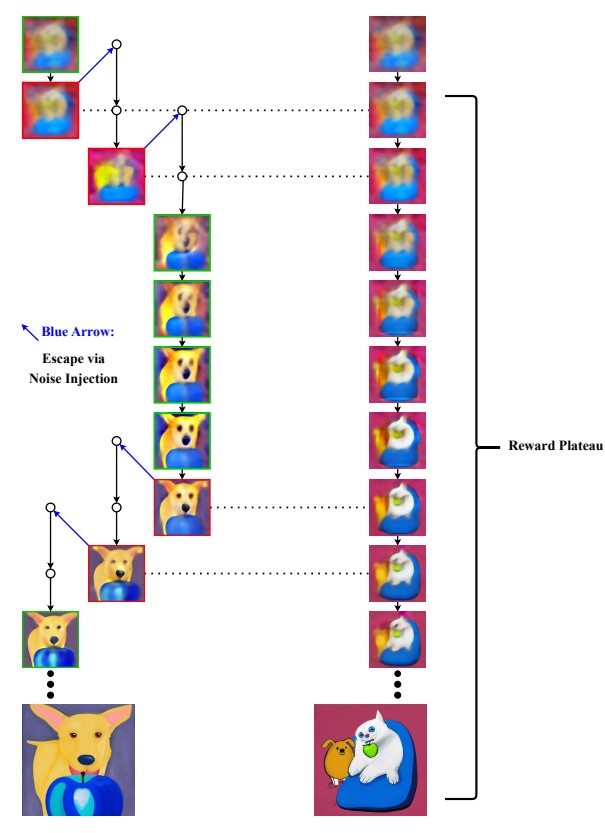

Figure 7: Sampling trajectories over the first 10 steps, comparing DDIM (right) and `Ctrl-Z` Sampling (left). Prompt: "a yellow dog and a blue apple". Red boxes mark steps where exploration is triggered, and green boxes mark regular steps. DDIM greedily steps towards higher probability regions and becomes trapped in a local optima state (white Dog). In contrast, `Ctrl-Z` Sampling explores via inversion to escape this local optima, converging to a higher-reward state (yellow Dog).

model already converges to reasonable generations in the middle steps, whereas later steps show only minor improvements that are barely perceptible. For red-highlighted steps, the updated $X_0^{t-1}$ after exploration is not visualized, and the subsequent $X_0^{t-2}$ shown in the image may still fail the reward check relative to the omitted updated state, thereby also being highlighted in red (these $X_0^{t-1}$ states are omitted in all figures for clarity).

`Ctrl-Z` Sampling improves sampling in two main ways. (1) Early-stage exploration. By injecting noise at the initial steps, `Ctrl-Z` Sampling perturbs low-frequency patterns and alters the denoising trajectory, effectively steering the model away from locally optimal states with limited future reward. For instance, in Fig. 8, explorations at steps 1–2 reduce the excessive blueness in the top-right corner of the dog and enhance the yellow tones in the center, yielding a more aligned trajectory. (2) Mid-stage refinement. Once low-frequency structures are largely fixed, `Ctrl-Z` Sampling continues to refine local details. As shown in Fig. 11, from step 15 onward it repeatedly improves alignment around the robot's eyes and hand pose, producing structurally consistent yet more plausible results.

It can also be observed that explorations near the 30th step often yield imperceptible changes, highlighting the limited effectiveness of late-stage explorations and motivating the use of an exploration window that restricts them to the first $\lambda$ steps. Nonetheless, some inefficiency remains. The reward-based adaptive initiation strategy is intended to suppress unnecessary explorations, yet when the image already reaches a satisfactory state early in the diffusion process, rewards tend to saturate, making further improvements difficult. This often triggers redundant explorations with trivial gains, as seen in Fig. 9, where many explorations were initiated despite negligible improvements in visual quality or alignment after the first 15 steps. These findings suggest that more sophisticated scheduling of exploration initiation could be a promising direction for future work.

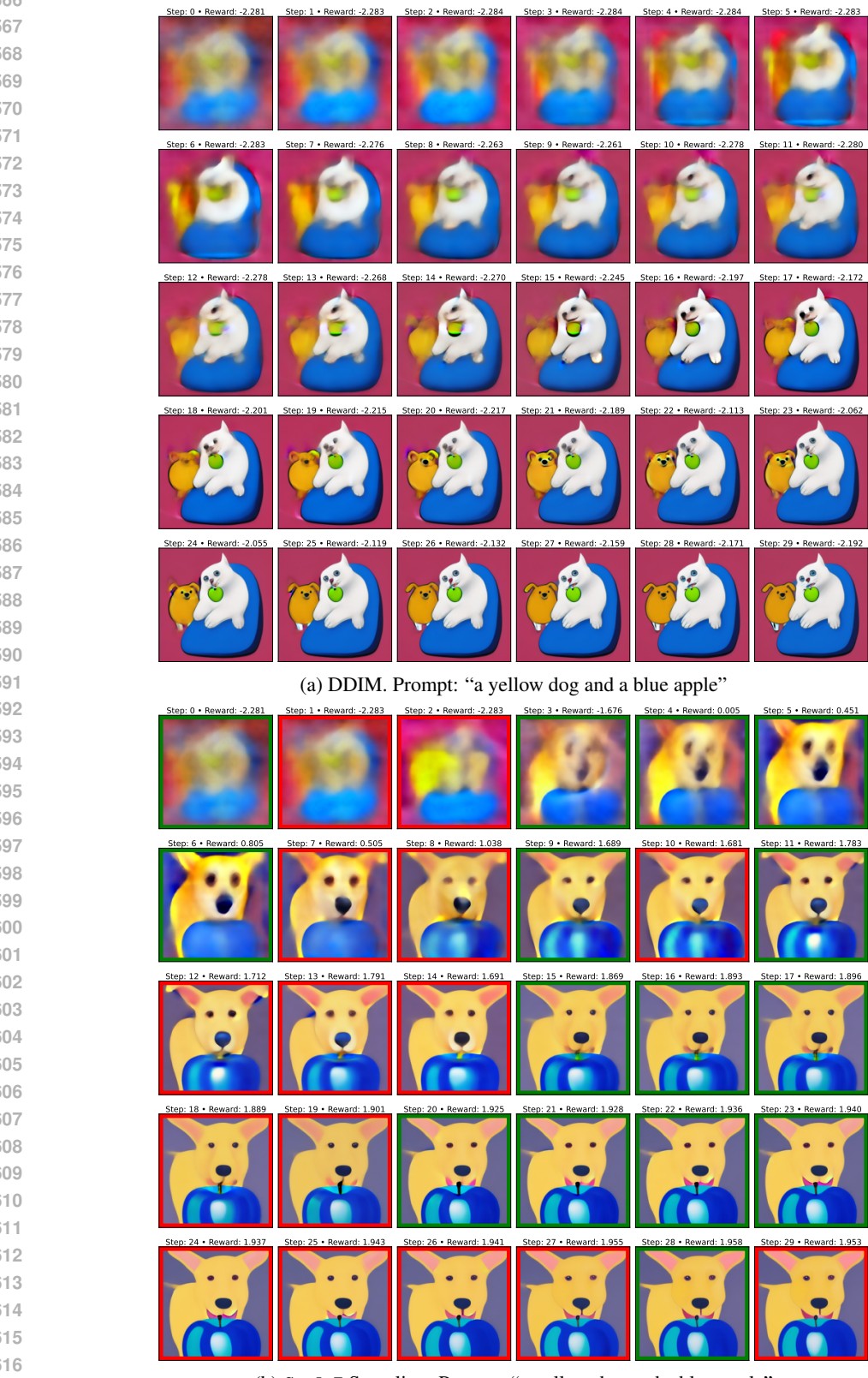

(a) DDIM. Prompt: "a yellow dog and a blue apple"

(b) `Ctrl-Z` Sampling. Prompt: "a yellow dog and a blue apple"

Figure 8: Qualitative comparison of the decoded $X_0^{t-1}$ across the first 30 over 50 generation steps. Steps with exploration initiated are highlighted in red, while others are shown in green.

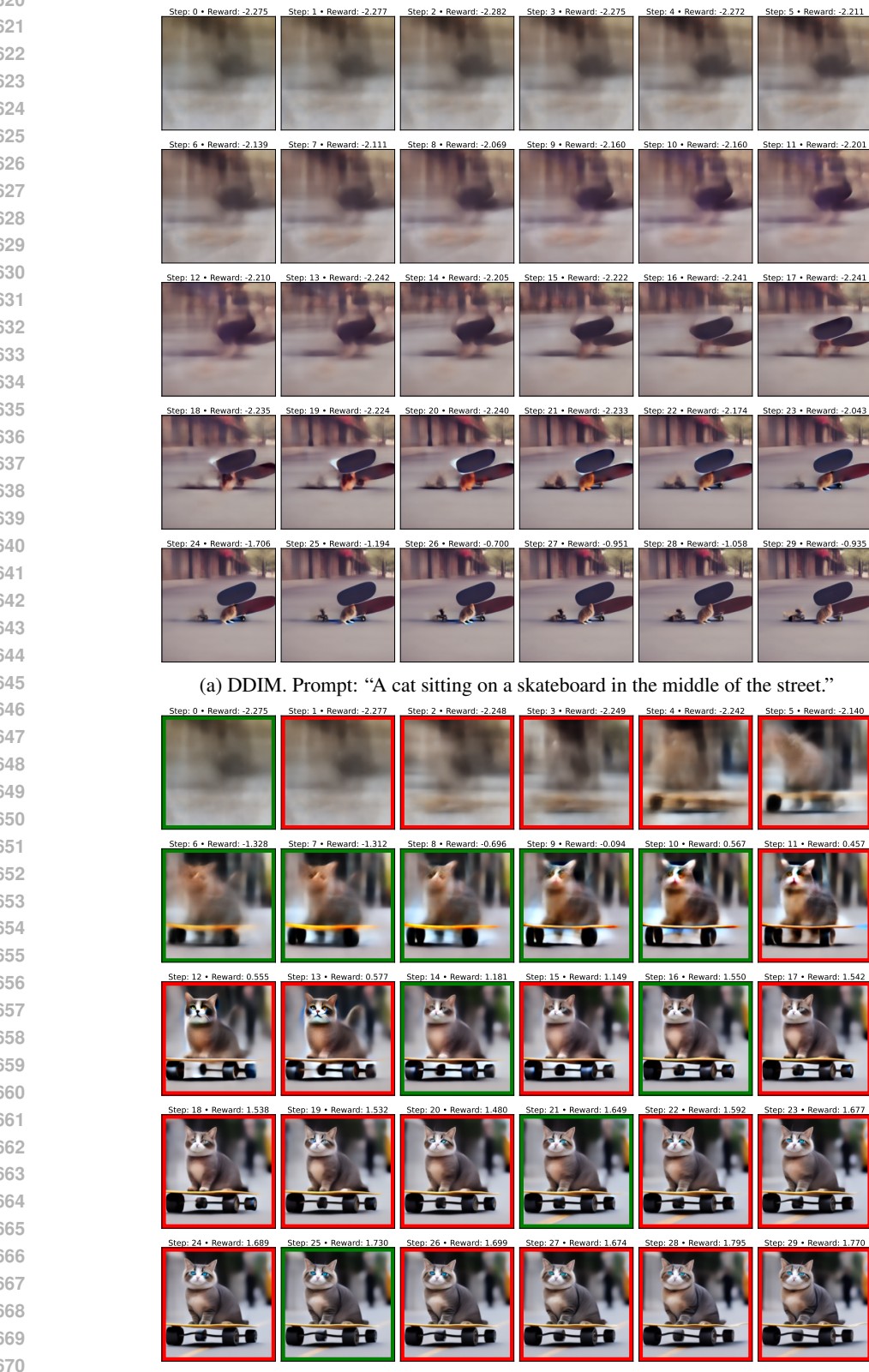

(a) DDIM. Prompt: "A cat sitting on a skateboard in the middle of the street."

(b) `Ctrl-Z` Sampling. Prompt: "A cat sitting on a skateboard in the middle of the street."

Figure 9: Qualitative comparison of the decoded $X_0^{t-1}$ across the first 30 over 50 generation steps. Steps with exploration initiated are highlighted in red, while others are shown in green.

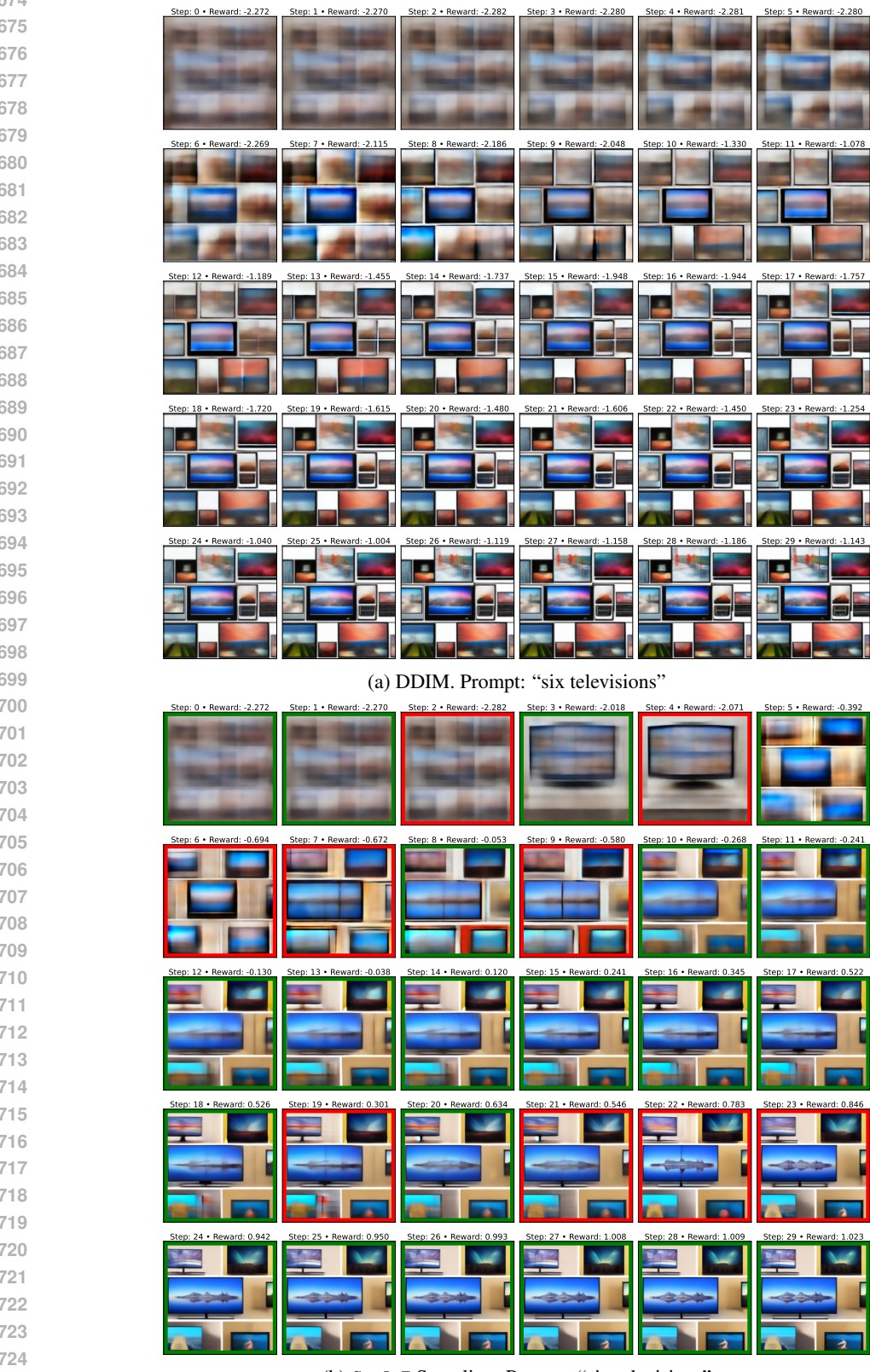

Figure 10: Qualitative comparison of the decoded $X_0^{t-1}$ across the first 30 over 50 generation steps. Steps with exploration initiated are highlighted in red, while others are shown in green.

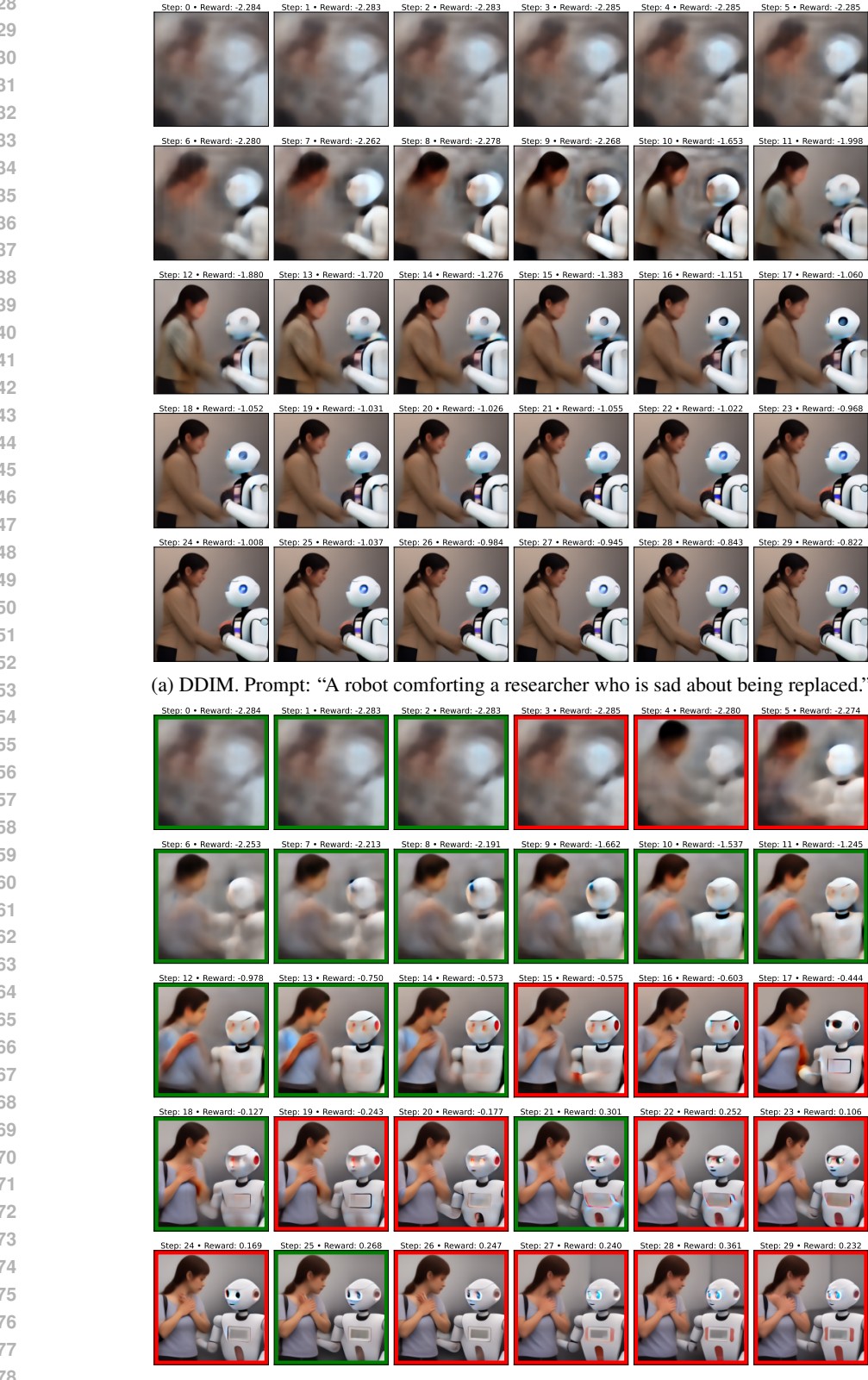

(a) DDIM. Prompt: "A robot comforting a researcher who is sad about being replaced."

(b) `Ctrl-Z` Sampling. Prompt: "A robot comforting a researcher who is sad about being replaced."

Figure 11: Qualitative comparison of the decoded $X_0^{t-1}$ across the first 30 over 50 generation steps. Steps with exploration initiated are highlighted in red, while others are shown in green.

