# OpenReview forum: "Ctrl-Z Sampling: Diffusion Sampling with Controlled Random Zigzag Explorations"
_ICLR.cc/2026/Conference — Submitted to ICLR 2026_

### Official Review · Reviewer_rfRj · 2025-10-25

**Soundness:** 2
**Presentation:** 3
**Contribution:** 2
**Rating:** 4
**Confidence:** 4

**Summary:**

The paper proposes a technique called "Control Z Sampling," which reframes the sampling process in conditional diffusion models as a form of hill climbing in a reward space. When the sampling process stagnates at a local optimum according to a reward model (e.g., ImageReward), the algorithm injects controlled noise and reverts to a noisier latent state to explore alternative trajectories. This process creates a backward-forward or "zigzag" exploration path intended to escape local optima. The method is presented as a model-agnostic plugin compatible with existing diffusion frameworks, requiring no retraining. Experiments show modest improvements in evaluation metrics compared to baselines. However, these gains come at a significant computational cost, requiring approximately 7.72 times the neural function evaluations (NFEs) on average while also requiring evaluation of a second reward model.

**Strengths:**

* The paper is well-written and mostly clear in its presentation.

* The idea of conceptualizing conditional diffusion sampling as a hill-climbing problem is interesting.

* The process of combining simulated annealing-like adaptive noise re-injection with a best-first search-like method to explore alternative trajectories is a reasonable blend of explore and exploit techniques.

* The quantitative and qualitative results clearly show that the method is having an effect, with moderate gains in performance metrics.

**Weaknesses:**

* The most significant weakness is the substantial computational cost, which averages roughly 7.72 times the neural function evaluations (NFEs) of the original process. In my opinion, this massive increase in energy and clock time makes the method practically unusable for what are only modest gains in output quality. Other hyperparameter choices for the algorithm would make this NFE multiplier even greater.

* Following up on the above point, the technique currently feels like a brute-force example or the first step in a research process, where one demonstrates how a search-based method *could* work but is then expected to introduce an efficient implementation to contrast with it. The paper has its heart in the right place but is essentially missing the efficient algorithm that would make the core concept viable.

* The fundamental re-framing of diffusion sampling as "search" or "hill climbing" is interesting but debatable, as diffusion models are fundamentally about sampling from a distribution, not necessarily about moving toward a single global optimum. A method that is too effective at achieving a global maximum could actually cause mode collapse by severely reducing output diversity.

* Related to the above point, the authors seem to be performing search in "reward space," but the structure of the diffusion sampling problem can easily lead a reader to think that the search is being performed in *probability* space. This distinction is not as clear as it could be in the text or Figure 1. Indeed, one comes away with opposite impressions from reading the abstract ("a form of hill climbing ... where the model iteratively refines a sample toward regions of higher **probability**") versus Section 4 ("as a hill-climbing process ... where each denoising step moves the sample toward regions of higher **reward**") [emphasis mine].

* The method primarily uses the ImageReward model, which is trained on clean images, to evaluate states at intermediate steps of the denoising process. This likely places the reward model's input out-of-distribution, which may impact the reliability of the guidance signal.

* The evaluation seems limited to latent space sampling and does not include testing in pixel space for computational reasons. This further limits the practical applicability of the work, as the field is increasingly moving back toward pixel-space diffusion.

* Some relevant work (e.g. Direct Noise Optimization [1], Diffusion Tree Sampling [2]) should be discussed in the context of the authors' method and ideally compared to if possible.

[1] Tang, et al (2024). Inference-Time Alignment of Diffusion Models with Direct Noise Optimization.

[2] Jain, et al. (2025). Diffusion Tree Sampling: Scalable inference-time alignment of diffusion models.

**Questions:**

1. The authors use a deterministic sampling process. Given that a significant amount of the benefits (exploring alternative paths, breaking out of local optima) could be inherently achieved more cheaply via stochastic sampling methods (Langevin Monte Carlo, DDPM), why was a comparison to or an evaluation on an established stochastic sampling process not included?

2. I would like to see the authors respond to the issues raised in the Weaknesses section.

**Details Of Ethics Concerns:**

No concerns.

---

> ### Author Response · Authors · 2025-11-17
> **Response to Weakness 1-6**
>
> We thank Reviewer rfRj for the detailed comments and discuss the weaknesses and the question below.
>
> ### **Weakness 1: Large NFEs Cost**
> As also discussed in our comments to all reviewers, inference scaling methods flexibility allows users to tune the hyper-parameters for different compute-quality trade-off in practice. We can still obtain better quality images with smaller compute, where we have updated the experiment results to reflect how Ctrl-Z remains effective under around 3X NFEs.
>
> ### **Weakness 2: More Efficient Algorithms**
> Related works (Ma et al., 2025...) in the inference scalaing domain and our methods does not aim to propose a zero-cost augmentation method，extra computational cost is a controlled and target-to-study component in the propose methods. Nonetheless, the reviewer propose an interesting problem domain of how to efficiently synchronize expensive inference scaling compute. This would be a orthogonal direction where methods like Diffusion Speculative Decoding (De Bortoli et al., 2025) focused on.
>
>
> ### **Weakness 3-4: Regarding Hill Climbing**
> We appreciate the reviewer’s concern and agree that diffusion models are fundamentally trained as generative samplers, not as optimizers of a single global objective. Our use of “hill climbing” is intended as an *inference-time, per-sample perspective* in a **sample quality space**, rather than a claim that we are globally maximizing a single mode of the model distribution. We have updated the paper to better reflect this.
>
> Concretely, each guided diffusion run starts from an independent noise sample and follows a path on a conceptual sample-quality landscape defined by a reward model $R$ (approximating semantic correctness, structural plausibility, etc.). Our “hill-climbing” view is strictly per-sample: we observe that some runs settle in local optima where the sample looks plausible but remains semantically flawed. Ctrl-Z explores around such states and only updates the trajectory if a branch achieves a strictly higher reward, but do not explores directly towards any higher-reward regions based on the reward model.
>
> This does not drive all runs toward a single global maximum. Different noise seeds still explore different basins of the quality landscape, and Ctrl-Z only refines them locally. In practice we do not observe reduced diversity across seeds—outputs remain diverse while their sample quality improves.
> Admittedly, the writing can be indeed misleading, we have updated the descriptions regarding the hill-climbing space as "sample quality space", which we estimate with a surrogate reward model.
>
>
> ### **Weakness 5: Distribution Shift in Reward Model Input**
> We agree that applying ImageReward to intermediate diffusion predictions introduces a mild distribution shift, since the model is trained on final “clean” images. In our method, however, the reward is computed on the decoded clean estimates $\hat{x}_0$, which already resemble low-resolution natural images rather than noisy latents. Empirically, the reward remains informative on such states: qualitative trajectories in the appendix show that ImageReward scores increase as intermediate images become more aligned with the prompt, and ablations with different reward models consistently yield gains in downstream metrics. Since Ctrl-Z only relies on relative ranking between nearby candidate states, it is robust to moderate miscalibration due to this shift, as reflected by improvements on final evaluation metrics. Developing reward models that are explicitly trained on (and efficient for) such intermediate states is an orthogonal but promising direction for future work.
>
>
> ### **Weakness 6: Extend to Image-Space Diffusion**
> Our experiments are conducted on latent diffusion models primarily for computational reasons. However, Ctrl-Z is not inherently tied to latent-space architectures: it only requires (i) a standard denoising update mapping $x_t \rightarrow x_{t-1}$, (ii) a rollback / noise-injection step that moves to a slightly noisier state before resuming denoising, and (iii) access at each step to a denoised prediction $\hat{x}_0^{(t)}$ (or equivalent clean reconstruction) that can be scored by a reward model. These ingredients are also available in common pixel-space diffusion implementations, where $x_t$ is an image tensor and the model’s prediction can be converted into an approximate clean image, so the same rollback-and-resample and reward-based local search can be applied in pixel space as well.
>
> The main limitation in *practical applicability* here comes from the backbone, not from Ctrl-Z. Recent pixel-space diffusion models, although demonstrate comparable performance to latent diffusion models, still incur substantially higher per-sample cost than latent diffusion. By contrast, latent diffusion is the workhorse in many practical systems precisely because of its efficiency, and our evaluation targets this widely used regime for practical applicability.

---

> ### Author Response · Authors · 2025-11-17
> **Response to Weakness 7 and Questions**
>
> ### **Weaknesss 7: Relevant Works**
>
> Direct Noise Optimization (DNO) focus on a different aspect of the generation process compared to Ctrl-Z Sampling. DNO treats the entire sampling process as a single, end-to-end, noise-to-sample mapping, $M_{\theta}(z)$. It then performs optimization via gradient ascent on the noise $z$ to find an input that maximizes a final reward function $r(M_{\theta}(z))$ for inference-time alignment, such gradient-based optimization can result in degenerated solutions and also be less efficient, compared to Ctrl-Z Sampling where reward model is only used to select from random renoised states.
>
> Diffusion Tree Sampling (DTS) is concurrent with our work and conceptually close to the concurrent method of Zhang et al. (2025), which we already discuss in the submission. Both DTS and Zhang et al. view diffusion sampling as a search problem and adopt DFS-style exploration to allocate computation to promising diffusion states. Ctrl-Z is also technically distinct from the search problem view: it implements a hybrid BFS/DFS strategy that first explores local neighbourhoods and only increases search depth when no better states can be found nearby. Empirically, these concurrent methods primarily study scaling behaviour in the regime of extremely large NFE budgets (e.g., $10^{2}$–$10^{5}$), whereas our focus is on a scalable sampler that delivers consistent gains under moderate, practically relevant test-time compute.
>
> We have updated the paper to clarify its relationship with the mentioned related works.
>
> ### **Question 1:**
>
> The reviewer is correct that stochastic samplers like DDPM explore, but their noise is a passive, local, and unguided part of the Markov chain.  In contrast, our Ctrl-Z noise is an active, guided, and adaptive escape mechanism. It is not part of the base sampler; it's an event triggered by reward stagnation. Such noise injection is not a built-in sampling step as in DDPM, which is why Ctrl-Z sampling could even be combined with DDPM.
>
> We deliberately chose the deterministic DDIM as our base for two critical reasons:
> 1. Reproducibility: It provides a fixed trajectory, ensuring any gains are exclusively from our Ctrl-Z mechanism.
> 2. Fairness: Our key baselines (SOP, Z-Sampling, Resampling) also build on DDIM. With scalable NFEs cost, DDIM is also more efficient for comparison.
>
>
>
> ### **References**
> Valentin De Bortoli, Alexandre Galashov, Arthur Gretton, Arnaud Doucet. Accelerated diffusion models via speculative sampling. In Forty-second International Conference on Machine Learning, 2025.
>
> Xiangcheng Zhang, Haowei Lin, Haotian Ye, James Zou, Jianzhu Ma, Yitao Liang, and Yilun Du. Inference-time scaling of diffusion models through classical search. arXiv preprint arXiv:2505.23614, 2025c.
>
> Nanye Ma, Shangyuan Tong, Haolin Jia, Hexiang Hu, Yu-Chuan Su, Mingda Zhang, Xuan Yang, Yandong Li, Tommi Jaakkola, Xuhui Jia, et al. Inference-time scaling for diffusion models beyond scaling denoising steps. arXiv preprint arXiv:2501.09732, 2025.

---

> > ### Comment · Reviewer_rfRj · 2025-11-17
> > **Response to Authors**
> >
> > I thank the authors for their thorough response to my review. In particular, I find the authors' take on the NFE issue, which I saw as the paper's main weakness, to be reasonable. I also believe that the revisions that the authors outlined will further improve the submission. This has relieved most of my concerns about the paper, and I will adjust my score accordingly.

---

> > > ### Author Response · Authors · 2025-11-18
> > >
> > > We sincerely thank the reviewer for their constructive discussion and for reconsidering the score.

---

### Official Review · Reviewer_J3qN · 2025-10-31

**Soundness:** 2
**Presentation:** 4
**Contribution:** 2
**Rating:** 2
**Confidence:** 3

**Summary:**

This paper proposes Ctrl-Z, a novel inference strategy for diffusion models to address semantic misalignment caused by convergence to local optima during conditional generation. The method leverages a reward-guided, controlled zigzag exploration process, dynamically alternating between forward refinement and backward steps with adaptive noise injection to escape optimization plateaus. Some experiments are provided to validate the effectiveness of the method.

**Strengths:**

* The approach is model-agnostic and substantially improves semantic alignment on text-to-image benchmarks, demonstrating a practical and efficient balance between exploration and computational cost.
* The writing and results are clear to the readers.
* Thorough ablation studies are conducted to show the robustness of the method.

**Weaknesses:**

* In the abstract, the authors state that denoising is analogous to climbing a probability hill, and that the process may get stuck at some local maxima. Why do you consider these local maxima suboptimal for generation? Could you visualize the hill and illustrate the imperfect samples corresponding to these local optima? My understanding is that if the score function has learned a distribution with local optimal points, then generating samples around these points is reasonable, since they belong to the true distribution. In that case, I do not quite understand the motivation for manually pushing the denoising process away from these local maxima.
* In Lines 43–46, you state that semantic misalignment and global inconsistency stem from local optima. Could you provide some evidence to support this claim?
* The computational cost increases to 7.72 times that of the original diffusion generation process, imposing a significant computational burden in industrial or commercial settings.
* You use ImageReward as the reward model during inference and also as the evaluation metric. This result is not convincing, since the evaluation metric is exactly what your method optimizes during generation. It is therefore expected that your method performs well on this metric. However, does this optimization degrade performance on other metrics? In addition, you do not report the FID score, which is the most commonly used metric in image generation. Although FID has some known limitations, I did not see any justification from the authors for omitting it.

**Questions:**

Please refer to Weaknesses.

---

> ### Author Response · Authors · 2025-11-17
> **Response to Reviewer J3qN**
>
> Thanks reviewer for the comments. We would like to clarify the confusions and questions below.
>
> ### **Weakness 1: Intuition of Ctrl-Z Sampling**
> **Why do you consider these local maxima suboptimal for generation?**
> When we call these states “suboptimal local maxima,” we do not claim that diffusion models converge to incorrect modes of the true data distribution. Rather, we refer to states where the denoising trajectory has reached a local optimum in a conceptual image quality space: the image is already locally plausible, but exhibits clear semantic flaws (e.g., wrong count, missing objects, broken spatial relations), and their reward (which is used as a surrogate for their quality in the conceptua quality space) plateau at values that are strictly lower than those achieved by nearby trajectories found by Ctrl-Z. Under this context, the states are suboptimal for the conditional generation objective we care about (in the conceptual quality space), even if they could lie in a high-density region of the learned data distribution. We have updated the paper to better describe the space the local maximas and hill-climbing refers to.
>
> **Visualizations**
> Directly visualizing the full hill landscape together with its local optima is challenging in such high-dimensional latent spaces. Instead, we provide a more concrete approximation in Figs. 8–11, where we visualize the pairs $(\hat{x}_0^{t-1}, R(c, \hat{x}_0^{t-1}))$ along the denoising trajectory. Although $\hat{x}_0^{t-1}$ is not the latent state itself, it is the predicted clean sample associated with that step and thus determines the prospective reward at that point. In these plots, reward plateaus can be clearly observed for DDIM (corresponding to local optima in our conceptual image quality space), while Ctrl-Z Sampling escapes these plateaus and reaches higher-reward states with visibly improved generations.
>
> **Why manually push away from them?**
> As clarified above, our “local optima” refer to optima in a **conceptual image quality space** that we approximate using reward models, rather than to modes of the probability distribution defined by the diffusion model. In practice, these local optima correspond to states where the sample and model has effectively converged and tends to stay, yet the resulting images remain globally unsatisfactory from the perspective of quality (e.g., wrong count, missing objects, incorrect relations). Ctrl-Z explicitly treats these as *quality-stagnant* regions and performs gradually enlarged local search in their neighbourhood, only moving away from the original trajectory when it finds an alternative with higher reward (and thus higher quality).
>
>
> ### **Weakness 2:**
> We empirically observe that steps identified as local optima by Eq. (5) correspond to points where denoising quality stagnates, and subsequent iterations yield little semantic improvement (see samples and their reward values in Appendix F). This aligns with our claim that semantic misalignment and global inconsistency stem from convergence to these suboptimal stable states.
>
> ### **Weakness 3:**
> While our adaptive exploration increases test-time compute, its design follows the inference-time scaling paradigm — trading additional compute for improved performance, similar to scaling trends in large language models and diffusion inference (Ma et al., 2025; Singhal et al., 2025). The 7.72× figure corresponds to the maximal exploration depth used for analysis; in practical deployments, we can adjust exploration depth and width to balance quality and cost. Even shallow exploration (≈2× DDIM cost) yields substantial gains, as shown in the updated results in Tab. 2. (See also the offical comments to all reviewers)
>
> ### **Weakness 4:**
> We do not optimize directly on one single evaluation metric but experiment with multiple metrics in addition to ImageReward. We also experiment with different reward models other than ImageReward. As reported in Tab. 3, improvements are **consistent** across different reward models and independent metrics, confirming that the method enhances general semantic alignment rather than overfitting to a specific reward function.
>
> **Why FID is infeasible**
> We do not report FID because our main benchmarks (e.g., Pick-a-Pic, DrawBench, T2I-CompBench) do not provide ground-truth images per prompt, making FID ill-defined in this open-world setting. Moreover, all compared methods share the same pretrained backbone and only differ in sampling strategy, so FID would mostly reflect the base model rather than the effect of our guidance. Instead, we follow recent work that emphasizes human-aligned metrics, which better capture the objectives of reward-guided inference. Prior works (e.g., He et al., 2025; Zhang et al., 2025c) use a similar set of metrics and exclude FID when studying inference-time scaling.

---

### Official Review · Reviewer_1wxx · 2025-11-01

**Soundness:** 2
**Presentation:** 3
**Contribution:** 2
**Rating:** 4
**Confidence:** 3

**Summary:**

The authors propose Ctrl-Z Sampling, a sampling strategy for diffusion models. The method targets a key limitation of standard denoising processes, which the authors liken to hill climbing: the tendency to become trapped in local optima. This can yield results that are semantically misaligned with the given condition or that lack global consistency. The core idea is an adaptive, reward-guided exploration mechanism. During sampling, a reward model monitors generation quality. If it detects optimization stagnation (suggesting a local trap), the method injects noise to roll back to a previous, noisier state. From that state, the reward model evaluates multiple candidate trajectories and selects a path that shows improvement. The rollback depth can be increased adaptively when shallow exploration fails. The authors report that this controlled “zigzag” exploration improves both generation quality and conditional alignment.

**Strengths:**

- The paper provides a clear and intuitive characterization of the problem, framing diffusion sampling as "latent space hill-climbing." This analogy effectively explains why samplers get stuck in local optima, leading to semantic mismatches or global inconsistencies. This narrative provides a unified motivational framework that helps readers grasp the necessity of the proposed strategy.
- The primary innovation lies in its feedback-controlled mechanism that determines when to explore and how deep the exploration should be. Unlike methods with fixed perturbation frequencies or amplitudes, Ctrl-Z Sampling uses reward stagnation as a trigger for exploration. It then progressively deepens the rollback (via DDIM-based noise injection) until a superior candidate trajectory is identified. This creates a controlled "forward-backward-forward" zigzag path.

**Weaknesses:**

- A core component of the method is the use of a reward model to detect stagnation ("Local Maxima Detection"). However, the paper does not propose a new reward model specifically designed or targeted for this task. Furthermore, the paper lacks discussion on several critical aspects of this component: 1) The stability and noise sensitivity of the chosen reward model. 2) The potential impact of reward misjudgments (false positives/negatives for stagnation) on the exploration path. 3) Any systematic biases in generation quality that might be introduced by the specific choice of reward model.
- The paper relies heavily on the "hill-climbing + rollback" intuition but provides no theoretical analysis to support it. Key guarantees are missing, such as convergence analysis, proof of escape from local optima, or bounds on the expected improvement. For instance, the paper does not provide any theoretical analysis of the relationship between the probability of successfully escaping a local optimum and the method's hyperparameters (e.g., the depth budget, the threshold $\delta$, the window $\lambda$).
- Although Ctrl-Z claims "controlled" exploration, its compute remains high: average NFEs are 7.72$\times$ (SD-2.1) and 8.79$\times$ (Hy-DiT) versus SOP’s 9.00$\times$, while quality gains over SOP are modest/inconsistent.

**Questions:**

Regarding the reward stagnation criterion, have you considered replacing the fixed threshold $\delta$ with a more flexible approach such as a dynamic or schedule-based threshold, a relative improvement test, or an adaptive gate based on reward variance or uncertainty? In regions where the reward values plateau at high scores, a constant $\delta$ might incorrectly interpret small but meaningful improvements as stagnation and trigger unnecessary rollbacks. How do you address this potential issue or prevent such misclassifications?

---

> ### Author Response · Authors · 2025-11-17
> **Response to Weakness**
>
> We thank Reviewer 1wxx for the detailed comments and address the three weaknesses below.
>
> **Reward model and local-maxima detection.**
> Our goal is not to introduce a new reward model, but to study how *adaptive exploration depth* can improve diffusion sampling under standard, off-the-shelf text–image reward models, in line with prior inference-time scaling work (Ma et al., 2025; Singhal et al., 2025; He et al., 2025; Zhang et al., 2025). As in these works, we treat reward-model design as orthogonal to our contribution and reuse existing verifiers. The inherent noise and instability of such reward models are mitigated by Ctrl-Z’s monotone update rule: at each step, we only accept candidates that improve the reward over the current best state, so occasional misjudgments tend to be averaged out over many updates rather than permanently locking the sampler into a poor state.
>
> Systematic bias from a *single* reward model is indeed unavoidable. To examine its impact, we run Ctrl-Z separately with different metrics used as the guiding reward and report all metrics in Tab. 3. We observe that (i) the largest gains typically appear on the metric that is used as the reward, but (ii) other alignment metrics also improve, albeit to a smaller extent. This indicates that, while different reward models have their own bias and scale, they are broadly aligned in preferring globally better image–text pairs: guiding the search with any of them still improves other metrics, rather than overfitting to a single scorer (For example, SOP achieves stronger performance only at IR compared to Ctrl-Z, demonstrating more bias from reward model).
>
>
> **“Hill-climbing + rollback” intuition and theory.**
> We agree that our analysis is primarily empirical. Our contribution is to show that an adaptive depth-first exploration scheme yields a better quality–compute trade-off than existing width-only methods under comparable NFE budgets, rather than to provide convergence guarantees. We partially validate the effectiveness of proposed method by ablating the core hyperparameters controlling exploration: $d_{\max}$, the width $N$, the acceptance threshold $\delta$, and the exploration window $\lambda$ in experiments, and observing stable, monotonic trends in reward and alignment metrics.
>
> We acknowledge that a rigorous analysis of escape probabilities or convergence for Ctrl-Z would be highly non-trivial. One would need to model (i) the learned denoiser/score network, (ii) the high-dimensional, non-convex reward landscape, and (iii) the non-Markovian, tree-structured sampling induced by hill-climbing + rollback. To our knowledge, even much simpler diffusion samplers lack such guarantees under realistic assumptions. We therefore view theoretical bounds as a technically demanding but valuable, *complementary* direction: our work aims to provide a clear empirical characterization of the behaviour of adaptive deeper exploration, which we hope can serve as a basis for future theoretical developments in this area. We would be grateful for any intuitions or pointers the reviewer may have toward such an analysis.
>
>
> **Controlled exploration and compute (NFEs).**
> As discussed in the offical comments, our notion of “controlled” refers to *controllable inference-time scaling*, not that Ctrl-Z is absolutely cheaper than SOP at a given target quality. Under comparable candidate settings, Ctrl-Z attains similar or better quality than SOP with slightly lower average NFEs (updated results demonstrates such superiority also at around 3X NFEs). Practitioners can tune $d_{\max}$, $N$, and $\lambda$ to trade off compute against quality, as illustrated by the scaling behavior in Fig. 2 and Tab. 6. We rephrase the claim in the paper to emphasize this budget-aware controllability rather than suggesting that the absolute compute is small.

---

> ### Author Response · Authors · 2025-11-17
> **Response to Questions**
>
> That's very insightful suggestion. Dynamic, schedule-based, or variance-aware thresholds are indeed promising directions, and similar ideas have already appeared in concurrent work (*e.g.*, Zhang et al., 2025). To keep our contribution conceptually clean and clearly distinguishable from these methods, we choose not to introduce an additional scheduling mechanism at this stage but leave them as future works, and instead focus on the effect of adaptive deeper exploration for inference scaling itself.
>
> Our stagnation test is deliberately simple: we use a fixed $\delta$, with $\delta = 0$ in the main settings. This choice has two advantages. First, it is broadly applicable across different reward models without requiring reward-specific tuning for scale or variance. Second, and directly related to the reviewer’s concern, with $\delta = 0$ Ctrl-Z never penalizes small but meaningful improvements: any strictly higher reward is accepted, regardless of its magnitude, so such steps are *not* interpreted as stagnation and do *not* trigger rollbacks. In other words, we trade sophistication for robustness and ease of use, while still obtaining a simple and effective stagnation signal.
>
> The effect of different $\delta$ values is empirically studied in App. Fig. 4. There we show that negative and positive $\delta$ values lead to different trade-offs between aggressiveness, robustness, and compute: more permissive (e.g., negative) $\delta$ encourages more aggressive exploration, whereas positive $\delta$ yields slightly more conservative updates at the cost of extra NFEs, often with somewhat higher quality.
>
> Most importantly, we do not position the stagnation criterion itself as the core contribution of Ctrl-Z. Our focus is on inference-time scaling via adaptive deeper exploration; the stagnation detector is a lightweight mechanism to make this exploration more cost-effective, analogous to pruning. Other inference-scaling methods such as SOP do not employ such scheduling mechanisms at all and still target the same core problem (scaling the diffusion inference process), which we see as the primary contribution space for our work.
>
>
> ### **References**
> Nanye Ma, Shangyuan Tong, Haolin Jia, Hexiang Hu, Yu-Chuan Su, Mingda Zhang, Xuan Yang, Yandong Li, Tommi Jaakkola, Xuhui Jia, et al. Inference-time scaling for diffusion models beyond scaling denoising steps. arXiv preprint arXiv:2501.09732, 2025.
>
>
> Raghav Singhal, Zachary Horvitz, Ryan Teehan, Mengye Ren, Zhou Yu, Kathleen McKeown, and Rajesh Ranganath. A general framework for inference-time scaling and steering of diffusion models. In International Conference on Machine Learning (ICML), 2025.
>
>
> Haoran He, Jiajun Liang, Xintao Wang, Pengfei Wan, Di Zhang, Kun Gai, and Ling Pan. Scaling image and video generation via test-time evolutionary search. arXiv preprint arXiv:2505.17618, 2025.
>
>
> Xiangcheng Zhang, Haowei Lin, Haotian Ye, James Zou, Jianzhu Ma, Yitao Liang, and Yilun Du. Inference-time scaling of diffusion models through classical search. arXiv preprint arXiv:2505.23614, 2025.

---

> > ### Comment · Reviewer_1wxx · 2025-11-26
> >
> > Thank you for the response. I see what you intend by treating the reward-model design and the stagnation criterion as orthogonal to your contribution, but I still have a few concerns. First, what exactly does “occasional misjudgments tend to be averaged out” mean in practice? With a monotone update rule, wouldn’t such errors be amplified rather than canceled? Second, regarding the “hill-climbing + rollback’’ intuition, the paper repeatedly relies on “hill-climbing + local maxima’’ language but provides no formal analysis, not even a toy example, so this comes across more as an informal narrative than a rigorous contribution. Lastly, while I appreciate the clarification that NFE is roughly 3× that of SOP, these results seem to suggest that Ctrl-Z does not offer a clear advantage over SOP-1.

---

> ### Author Response · Authors · 2025-11-27
>
> We appreciate the insightful comments from the reviewer. We make further clarifications below:
>
> ### **Q1**
> In unconstrained optimization over a static landscape, a monotone update rule on a noisy objective can indeed "lock" the search into false maxima. However, we clarify that in our framework, it will not be a critical concern for the following reasons:
>
> **The Diffusion Prior as a Manifold Constraint:**
> Crucially, Ctrl-Z does not perform unbounded optimization on the image space. Instead, it selects from a finite set of candidates generated by the pretrained denoiser $\Phi$. This distinction is vital: reward models are mostly confounded by high-frequency noise that lies off the manifold of diffusion models. In our method, the reward model $R$ never encounters such "glitchy" states because the candidate pool is already strictly constrained by the diffusion prior itself. The diffusion prior dominates over any reward-based searches.
> While the reward model can still inherently possess some biases distinct from the diffusion model, this is an intentional feature. Our approach rests on the premise that **"judging is easier than generating."** Since the base diffusion distribution is suboptimal, a reward model can identify and select superior states from the valid candidates. Thus, rather than amplifying error, the reward model acts to refine the diffusion process, steering the trajectory towards higher-quality regions of the manifold.
>
> **Non-Stationary Optimization Landscape:**
> The raised concern about error amplification assumes a static optimization surface where a false peak persists throughout the search. However, **diffusion sampling is a non-stationary process**. The landscape of reachable candidates changes at every timestep $t$ as the noise level and distribution shift.
> A specific bias or a false local maximum in the reward model at step $t$ is often transient: it relies on specific noise artifacts that disappear or change form at step $t-1$. Consequently, a biased selection at one step does not permanently degrade the trajectory. The model encounters a fresh set of candidates at the next exploration step, allowing judging errors to **"average out"** over the trajectory. As long as the reward model does not systematically prefer failures across different states, the sequential diffusion process allows the model to recover from individual misjudgments.
>
> **Empirical Validation:**
> This robustness is supported by our experiment in Fig. 2,4. We observe that increasing the exploration depth and number of candidates consistently improves other metrics alongside the guiding reward (ImageReward). If reward bias were accumulating, we would expect the guiding reward to increase while the other metrics decrease. The observed positive correlation confirms that the method effectively leverages the reward model without overfitting to its occasional bias.
>
>
>
> ### **Q2**
> We agree that the initial "hill-climbing" analogy benefited from a more formal definition. In the revised manuscript (Sec 4), we have rewritten the problem definition to explicitly frame the sample quality space as the optimization landscape, with the reward model serving as its surrogate. In this view, standard diffusion trajectories represent greedy optimization paths on the reward landscape. We have also updated the contribution statements to avoid overclaiming.
>
> To address the request for a toy example demonstrating the hill climbing intuition, we refer the reviewer to the trajectory analysis in Fig. 7. This empirically maps the landscape for the prompt "a yellow dog and a blue apple":
>
> - **Local Optima.** DDIM greedily steps towards higher probability regions and converges to a "White Dog" state. This represents a local optimal where the image is plausible (high prior probability) but misaligned (low reward), and the greedy path lacks the mechanism to escape.
>
> - **Backtracking & Perturbation.** Ctrl-Z detects the reward plateau and triggers inversion. This noise inversion step expands the search radius, forcing the trajectory away from the current local optimal.
>
> - **Transition to Better Optima.** This exploration allows the exploration to discover and re-converge into a distinct, higher-reward state (the "Yellow Dog") that was inaccessible to the initial greedy path.
>
>
> ### **Q3**
> With comparable NFE budgets, Ctrl-Z consistently outperforms SOP on HPSv2 and PickScore, two human-aligned preference metrics that directly approximate human judgments of conditional image quality, similar to ImageReward. In cases where SOP attains slightly higher ImageReward, this mainly reflects more aggressive local search around the ImageReward signal and tends to overfit that specific reward model, without improving HPSv2/PickScore as much (as discussed in the paper). We also update the results on T2I-CompBench, and further demonstrate the advantages of Ctrl-Z Sampling: it achieves consistently stronger scores than SOP across compositional sub-tasks.

---

### Official Review · Reviewer_Lfgi · 2025-11-03

**Soundness:** 3
**Presentation:** 2
**Contribution:** 2
**Rating:** 4
**Confidence:** 4

**Summary:**

This paper proposes Ctrl-Z sampling, a sampling method for diffusion models designed to improve output quality by escaping local optima during the denoising process. The approach evaluates the generated state after each denoising step to determine whether it is a local optimum; if so, it injects an adaptive amount of noise to revert the process to a noisier state. This backward exploration mechanism is intended to leads to better text alignment and enhanced visual quality, while costing approximately 7.72× more function evaluations (NFEs) compared to standard diffusion sampling. The effectiveness of the proposed method is demonstrated through experiments on standard text-to-image benchmarks.

**Strengths:**

* Interesting problem: The paper addresses an interesting and relevant issue in diffusion models — the tendency of sampling trajectories to stagnate or over-converge — which could be impactful.
* Extensive empirical evaluation: The paper includes a comprehensive set of experiments across multiple text-to-image benchmarks, providing both quantitative metrics and qualitative visualizations.
* Thoroughness and effort: The work is detailed and carefully executed, with significant experimental effort and a long appendix that documents settings, ablations, and implementation details.

**Weaknesses:**

- The problem studied in this paper is not well explained or sufficiently justified. From the outset, in both the abstract and introduction, the authors describe how the denoising process converges to local optima. For example, lines 43–46 state:

      “Despite their strong generative performance, diffusion models often exhibit semantic misalignment or global inconsistency in conditional generation. These issues arise when the denoising process converges to local optima that prioritize local visual plausibility over semantic relevance or structural coherence.”

    The main concern is that it is unclear what the term *local optima* means in the context of diffusion models, as no explicit optimization occurs during inference. The authors could have more effectively explained and motivated this concept by framing it as follows: the denoising process is an iterative refinement procedure in which samples may occasionally collapse to suboptimal regions of the data manifold (for example, producing blurry or incomplete images). This underlying phenomenon of sampling stagnation or collapse could then be described as becoming stuck in a local optimum.


- The procedure for detecting local optima during sampling is not well justified. The paper lacks an intuitive explanation or illustrative experiment demonstrating why the criterion defined in Equation (5) effectively identifies states corresponding to local optima.

- The experimental results are not particularly convincing. The quantitative improvements over the baselines, especially on the SOP dataset, are marginal and do not clearly demonstrate a significant advantage. While the qualitative examples in Figure 3 more effectively illustrate the benefits of Ctrl-Z sampling, the additional qualitative results provided in the supplementary material appear less compelling and more comparable to benchmarks.

- Considering the quality of the results, I am concerned that the approximately 7.72× increase in NFEs may not justify the relatively modest gains reported in the experiments.

**Questions:**

1. The definition of $\Phi$ provided in line 139 and equation 1 conflict as the output of $\Phi$ is $x_0$.
2. Line 153, it's better if "clean image" was replaced with "clean sample" as image was never used up to this point.
3. Line 157: Why is the sample estimate $\hat{x_0}$ a good proxy for the final clean sample $x_0$?
4. Line 161: The concept of "inversion" is not a very popular concept. It would be useful to add an explicit description of inversion by something as short as "...inversion (re-noising)...".
5. Line 181: "forward denoising" is probably incorrect, right? In the forward process, noise is added and in the backward, noise is removed.
6. Is the "Resampling" method discussed in Figure 1 the same as the stochastic SDE sampler?
7. Line 210: "Zigzag Sampling takes a backward step along the direction of un-
conditional generation, alternating between conditional and unconditional denoising to better inject
conditioning signals." --> what does conditioning refer to here? condition on what exactly?
8. Line 223: It would be beneficial to clarify what is meant by "latent trajectory" as it's not a standard term used in the diffusion community.
9. Why does the criteria of equation 5 make sense?

---

> ### Author Response · Authors · 2025-11-17
> **Response to Questions**
>
> Thanks Review Lfgi for the detailed reading and insightful suggestions, I would like to first answer all the questions raised in the review, then make a few clarifications regarding the argued Weakness for this paper.
>
> ### **Response to Questions**:
>
> 1.	The differences here is that $\Phi$ in L139 is in global form describing the whole process, whereas $\Phi^t$ in Eq.1 address the per-step operation. Similar definitions and usage of such notations can be found in Bai et al. (2024).
> 2.	Paper have been updated accordingly.
> 3.	Diffusion models are trained so that, for any timestep $t$, the predicted noise $\epsilon_\theta^t(x_t, c)$ allows reconstructing the underlying clean sample via:
> $$
> \hat{x_0}(x_t, c) = \frac{x_t - \sqrt{1 - \alpha_t}\,\epsilon_\theta^t(x_t, c)}{\sqrt{\alpha_t}}.
> $$
> In DDIM sampling , the next latent is then a convex combination of this clean estimate and the same predicted noise:
> $$
> \begin{aligned}
> x_{t-1} = \Phi^t(x_t, c) &=
> \sqrt{\alpha_{t-1}} \cdot \frac{x_t - \sqrt{1 - \alpha_t} \cdot \epsilon_\theta^t(x_t, c)}{\sqrt{\alpha_t}}  + \sqrt{1 - \alpha_{t-1}} \cdot \epsilon_\theta^t(x_t, c).
> \end{aligned}
> $$
> At late denoising steps, $\sqrt{1-\alpha_{t-1}}$ becomes small, so $x_{t-1}$, and consequently the final $x_0$, is dominated by $\hat{x}_0$. This is precisely why DDIM can sample efficiently with a small number of discrete steps, and it justifies using $\hat{x}_0$ as a proxy for the final clean sample. Empirically, as shown in Figs. 8-11, decoding $\hat{x}_0$ at $t = 30/50$ already yields high-resolution, semantically reasonable images, further supporting this choice.
> 4.	Paper have been updated accordingly.
> 5.	The usage of word ‘forward’ is reasonable given that we have defined the operation of re-noising the latent as an ‘inversion’ process (going backward in the denoising path), hence denoising would use the opposite: ‘forward’.
> 6.	There is a slight difference. The “Resampling” baseline follows the idea of Lugmayr et al. (2022): at a chosen step we briefly revert to a noisier latent and then continue with standard DDIM denoising. Concretely, we apply our inversion operator Ψ from Eq. (4) with a small ∆ (in our implementation, ∆ = 1),
> $$
> x_{t+\Delta} = \Psi\bigl(x_t, \Delta; \epsilon\bigr)
> $$
> where $\epsilon \sim \mathcal{N}(0, I)$. This inverse update moves the latent to a previous (noisier) timestep $t+\Delta$ on the DDIM schedule, so recovering $x_{t-1}$ requires $\Delta+1$ additional deterministic DDIM updates $\Phi_k$ applied from $x_{t+\Delta}$ back to $x_{t-1}$. Thus, *Resampling* performs a single local ''rollback via noise injection + extra denoising'' within an otherwise deterministic DDIM trajectory.
> In contrast, the stochastic SDE sampler corresponds to simulating the reverse-time SDE, where Gaussian noise is injected at every reverse step and the entire trajectory is governed by the SDE discretization rather than DDIM. We do not use that sampler in this work.
> 7.  L210 describes the Z-sampling method by Bai et al. (2024). Conditioning refers to *Conditional Generation* $\epsilon_\theta(x_t, \varnothing)$: Denoising guided by the text prompt (and also CFG). And *Unconditional Generation* $\epsilon_\theta(x_t, c)$: Denoising without any prompt guidance. We admit the statement is ambigous and have updated it in the revised paper.
> 8.  In our paper, we use *latent trajectory* to refer to the sequence of latent states $\{x_t\}$ produced by the model during the denoising process. Since we also discuss the *reward trajectory* (the sequence of reward values $\{r_t\}$ used to detect stagnation), we update the paper to explicitly define both terms at their first occurrence and use them consistently throughout the revised manuscript.
> 9.  See response to Weakness 2
>
> ### References
> Andreas Lugmayr, Martin Danelljan, Andres Romero, Fisher Yu, Radu Timofte, and Luc Van Gool. Repaint: Inpainting using denoising diffusion probabilistic models. In Proceedings of the IEEE/CVF conference on computer vision and pattern recognition (CVPR), pp. 11461–11471, 2022.
>
> Lichen Bai, Shitong Shao, Zikai Zhou, Zipeng Qi, Zhiqiang Xu, Haoyi Xiong, and Zeke Xie. Zigzag diffusion sampling: Diffusion models can self-improve via self-reflection. In Proceedings of the International Conference on Learning Representations (ICLR), 2024.

---

> ### Author Response · Authors · 2025-11-17
> **Response to Weakness**
>
> ### Weakness 1:
> We agree with the first point that our use of “local optima” can be phrased more clearly and decently as sampling stagnation or collapse to suboptimal regions of the data manifold, we appreciate the reviewer for the efforts on this and we adopt this framing in the revised paper.
>
> ### Weakness 2:
> Having explained why $\hat{x}_0$ is a good estimate of $x_0$, Eq. (5) follows from a simple assumption: along a standard DDIM trajectory, each denoising step removes part of the uncertainty noise from the latent and thus tends to move $\hat{x}_0^{t-1}$ toward a state of *higher quality* than earlier steps. With a reward model as a surrogate function to estimate such conceptual quality of the sample, the reward sequence
> $$
> r_t = R(c, \hat{x}_0^{t-1}),
> $$
> can be viewed as a hill-climbing curve along the denoising trajectory in the quality space. Eq. (5) then defines a “local optimum” in this *trajectory-wise* sense: we trigger exploration when $r_t$ fails to exceed the best reward seen so far by a small margin $\delta$, indicating that the trajectory has entered a plateau where further DDIM steps mainly refine details without improving alignment.
> We empirically validate that this criterion is meaningful rather than arbitrary. In Sec. C.3 (Tabs. 5–6), the reward-based triggering in Eq. (5) matches or surpasses “always explore” in quality while using substantially fewer NFEs, and clearly outperforms random triggering under similar compute. In Sec. F, we visualize the reward, the trigger points, and the corresponding images: triggers consistently occur where the reward flattens and the images look plausible but are still misaligned (e.g., wrong counts or colors), and the subsequent Ctrl-Z explorations move the trajectory to states with higher reward and better visual agreement with the prompt (especially visually observable in early denoising steps).
>
> ### Weakness 3-4:
> Regarding points (3) and (4), our main goal is to study inference-time scaling, not to propose a zero-cost sampler (See also the offical comments to all reviewers). Both SOP and the proposed Ctrl-Z Sampling are mechanisms for *flexibly and adaptively* increasing NFEs at test time. Across our main benchmarks, Ctrl-Z consistently matches or improves over the SOP baseline under comparable or smaller NFE budgets, and these gains appear across multiple metrics.
> The configuration with roughly $7.72\times$ the original NFEs is chosen as a representative scaling regime to compare Ctrl-Z against SOP under a moderate compute budget; it is not an inherent requirement of our method. As shown in Fig. 2, Ctrl-Z exposes a smooth compute–quality trade-off: by adjusting parameters such as the maximum inversion depth $D_{\max}$ and the number of candidates. We have also added the performance of Ctrl-Z and SOP under fewer NFEs to reflect its performance gain with flexibility.

---

### Author Response · Authors · 2025-11-17
**Official Comment: Ctrl-Z Sampling as a scalable inference-time method**

We thank all reviewers for their careful reading and constructive feedback. We would like to clarify a central aspect of the contribution that may not have been sufficiently emphasized in the current draft.


Several critiques focus on the *absolute* improvement achieved at around $7\text{–}8\times$ the NFEs of standard DDIM, interpreting our method primarily as a fixed-cost sampler with limited gains. Our intention, however, is to position Ctrl-Z Sampling as a low-cost **inference-time scaling mechanism** for diffusion models, rather than optimizing a single fixed-cost sampler. Inference-time scaling, which spends more test-time compute to systematically improve performance, has become a key paradigm in NLP and LLMs, where deeper decoding, reranking, and test-time optimization reliably trade extra compute for better outputs. In contrast, diffusion models have seen far fewer principled approaches in this direction; most prior efforts focus on architecture or training-scale changes to improve diffusion model, or more efficient sampling strategies during inference, rather than exposing controllable test-time scaling side of diffusion, aside from increasing sampling steps (which is of limited effectiveness).

Ctrl-Z sampling exposes explicit controls over exploration depth $d_{\max}$, width $N$, and the exploration window $\lambda$, allowing users to trade off additional test-time compute for more performance gains, in line with recent work on inference/test-time scaling in diffusion and language models (Ma et al., 2025; Singhal et al., 2025; He et al., 2025; Zhang et al., 2025). However, these works primarily characterize scaling laws under extremely large NFE budgets (e.g., $10^{2}$–$10^{5}$), whereas our contribution is a scalable sampler that remains effective under moderate, practical test-time compute.

In the main tables we chose a mid-range configuration (≈$7\text{–}9\times$ NFEs) so that Ctrl-Z can be fairly compared to SOP under similar compute budgets. However, the method is not tied to this single operating point. As shown in our ablations (Fig. 2 and Table 6), increasing exploration depth and/or the number of candidates yields consistent improvements across alignment metrics (HPSv2, PickScore, ImageReward), with AES being naturally less sensitive because it does not directly assess conditioning alignment. In other words, Ctrl-Z defines a *family* of samplers along a compute–quality curve, rather than a single fixed sampler.

By contrast, the baseline “self-refinement” methods we compare against (Resampling and Z-Sampling) offer only shallow, fixed-strength perturbations. In our experiments, simply stacking more resampling or zigzag iterations does not scale their performance with NFEs and can even degrade quality. SOP is the most direct inference-time scaling baseline; under comparable NFE budgets, Ctrl-Z matches or outperforms SOP on human-aligned metrics while using fewer or similar NFEs, and additionally benefits from *depth* as an extra scaling axis rather than relying solely on wider local search.

To avoid this misunderstanding, we revise the title to explicitly frame Ctrl-Z Sampling as an scalable inference-time method and to make the compute–quality trade-off and scaling behavior more prominent in the exposition. We also provide how Ctrl-Z and SOP sampling method performs with lower inference budget similar to Resampling and Z-Sampling.




### **References**
Nanye Ma, Shangyuan Tong, Haolin Jia, Hexiang Hu, Yu-Chuan Su, Mingda Zhang, Xuan Yang, Yandong Li, Tommi Jaakkola, Xuhui Jia, et al. Inference-time scaling for diffusion models beyond scaling denoising steps. arXiv preprint arXiv:2501.09732, 2025.


Raghav Singhal, Zachary Horvitz, Ryan Teehan, Mengye Ren, Zhou Yu, Kathleen McKeown, and Rajesh Ranganath. A general framework for inference-time scaling and steering of diffusion models. In International Conference on Machine Learning (ICML), 2025.


Haoran He, Jiajun Liang, Xintao Wang, Pengfei Wan, Di Zhang, Kun Gai, and Ling Pan. Scaling image and video generation via test-time evolutionary search. arXiv preprint arXiv:2505.17618, 2025.


Xiangcheng Zhang, Haowei Lin, Haotian Ye, James Zou, Jianzhu Ma, Yitao Liang, and Yilun Du. Inference-time scaling of diffusion models through classical search. arXiv preprint arXiv:2505.23614, 2025.

---

### Meta-Review · Area_Chair_wtUo · 2026-01-11

**Summary:**

The paper proposes a reward-guided diffusion sampling strategy for conditional generation that aims to escape local optima during denoising. The reviewers found that the paper is well-written and mostly clear, addresses a relevant problem, that the main hill-climbing framing is clear (though questioned by some reviewers), and that the empirical evaluation is extensive with thorough ablation studies. However, the reviewers also raised significant concerns about an unclear definition of "local optima", limited theoretical analysis (e.g., convergence), and limited gains relative to the substantial additional computational cost.

**Reviewer Concerns:**

The rebuttal addresses concerns about "local optima"; however, the concerns about computation cost and theoretical analysis remain.

**Reviewer Scores:**

The score of Reviewer 1wxx would likely remain unchanged, as the reviewer's main concerns (cost and theory) were not addressed. The scores of Reviewers Lfgi and J3qN would likely remain similar or slightly increased, as some of their concerns were addressed. The score of reviewer rfRj would likely increase as most of the reviewer's concerns were addressed. However, given the initial scores, it is unlikely that the overall evaluation would shift sufficiently toward acceptance, even if some reviewers were to increase their scores.

---

### Decision · Program_Chairs · 2026-01-26

Reject